



# Novel Pathway of SO₂ Oxidation in the Atmosphere: Reactions with Monoterpene Ozonolysis Intermediates and Secondary Organic Aerosol

Jianhuai Ye[1], Jonathan P. D. Abbatt[2], Arthur W.H. Chan[1]

[1]Department of Chemical Engineering & Applied Chemistry, University of Toronto, Toronto, Canada
[2]Deparment of Chemistry, University of Toronto, Toronto, Canada

*Correspondence to*: Arthur W.H. Chan (arthurwh.chan@utoronto.ca)

**Abstract.** Ozonolysis of monoterpenes is an important source of atmospheric biogenic secondary organic aerosol (BSOA). While enhanced BSOA formation has been associated with sulfate-rich conditions, the underlying mechanisms remain poorly understood. In this work, the interactions between SO₂ and reactive intermediates from monoterpene ozonolysis were investigated under different humidity conditions (10% vs. 50%). Chamber experiments were conducted with ozonolysis of α-pinene or limonene in the presence of SO₂. Limonene SOA formation was enhanced in the presence of SO₂, while no significant changes in SOA yields were observed during α-pinene ozonolysis. Under dry conditions, SO₂ primarily reacted with stabilized Criegee Intermediates (sCI) produced from ozonolysis, but at 50% RH, heterogeneous uptake of SO₂ onto organic aerosol was found to be the dominant sink of SO₂, likely owing to reactions between SO₂ and organic peroxides. This SO₂ loss mechanism to organic peroxides in SOA has not previously been identified in experimental chamber study. Organosulfates were detected and identified using electrospray ionization-ion mobility time of flight mass spectrometer (ESI-IMS-TOF) when SO₂ was present in the experiments. Our results demonstrate the synergistic effects between BSOA formation and SO₂ oxidation through sCI chemistry and SO₂ uptake onto organic aerosol and illustrate the importance of considering the chemistry of organic and sulfur-containing compounds holistically to properly account for their reactive sinks.





## 1. Introduction

Secondary organic aerosol (SOA) is formed from condensation of low-volatility products from atmospheric oxidation of volatile organic compounds (VOCs) and comprises a major fraction of atmospheric organic aerosol (Jimenez et al., 2009). Globally, the dominant fraction of SOA is formed from oxidation of biogenic precursors, as suggested by the high fractions of modern carbon in atmospheric organic aerosol (Goldstein et al., 2009; Weber et al., 2007; Szidat et al., 2006; de Gouw et al., 2005). While emissions of biogenic hydrocarbons are largely uncontrollable, laboratory studies and field observations have shown that biogenic SOA (BSOA) formation is influenced by anthropogenic emissions, such as primary organic aerosol and $NO_X$ (Ye et al., 2016; Xu et al., 2015; Goldstein et al., 2009; Ng et al., 2007, 2008). As a result, it has been suggested that atmospheric BSOA could be significantly reduced by controlling anthropogenic pollutants (Carlton et al., 2010; Heald et al., 2008).

One important pollutant that can affect BSOA formation is $SO_2$, with up to 94% of its emissions from anthropogenic activities such as fuel combustion in the U.S. (Year 2014; U.S. EPA, 2014) and more than 78% globally (Year 2007-2009; McLinden et al., 2016). Oxidation of $SO_2$ in the atmosphere leads to formation of sulfuric acid that plays a crucial role in atmospheric new particle formation (Brock et al., 2002) and enhances SOA formation through acid-catalyzed mechanisms (Jang et al., 2002). Long-term ground observations in Southeast U.S. show that the decrease of BSOA is correlated strongly to the decrease in sulfate content in aerosols (Marais et al., 2017), implying co-benefits in controlling $SO_2$ emission to reduce both sulfate and BSOA. It is further demonstrated by Xu et al. (2015) that anthropogenic $NO_X$ and sulfate correlate strongly with 43-70% of total measured organic aerosol in this area. The mechanisms by which sulfate influences BSOA formation have also been demonstrated through laboratory studies. For example, SOA yields of isoprene, as well as α-pinene and limonene, increased with increasing the acidity of the seed aerosol (Iinuma et al., 2007; Surratt et al., 2007; Gao et al., 2004; Czoschke et al., 2003). The formation of high-molecular-weight (high-MW) oligomers and organosulfates is enhanced in the presence of sulfuric acid (Surratt et al., 2008; Tolocka et al., 2004).





There is also increasing evidence that $SO_2$ may directly influence BSOA formation by influencing OH reactivity. In the presence of $SO_2$, enhanced gas-phase products from α-pinene and β-pinene photooxidation were observed with a decreased oxidation state of gas-phase semivolatile species (Friedman et al., 2016). Liu et al. (2017) demonstrated that SOA yields of cyclohexene photooxidation were lower at atmospherically relevant concentrations of $SO_2$, implying that $SO_2$ may indirectly decrease SOA formation when the acid-catalyzed SOA enhancement is insufficient to compensate for the loss of OH reactivity towards VOCs. $SO_2$ can also directly influence VOC oxidation mechanisms through reactions with stabilized Criegee intermediates (sCI) formed from olefin ozonolysis (Huang et al., 2015a; Welz et al., 2012). Field observations suggested that $SO_2$ + sCIs reactions may contribute up to 50% of the total gaseous sulfuric acid production in the forest atmosphere, which is comparable to that from gas-phase oxidation by OH (Mauldin III et al., 2012). Consistent with this observation, model calculations with $CH_2OO$, the simplest sCI, suggested that the $SO_2$ and sCI reaction could be significant in atmospheric sulfuric acid formation under dry conditions, but suggest that this pathway may become less important as humidity increases due to the scavenging effect of water and water dimer towards sCIs (Calvert and Stockwell, 1983). However, the reactivity of sCI towards $SO_2$ is observed to be strongly dependent on its molecular structure. While $CH_2OO$ may primarily react with water and water dimer, Huang et al. (2015) demonstrated that the di-substituted sCI has a long lifetime under atmospherically relevant humidity conditions and may react with $SO_2$. Sipilä et al. (2014) also demonstrated that the formation rates of sulfuric acid, the product of the sCI + $SO_2$ reaction, are nearly independent of humidity for monoterpene ozonolysis. In addition to OH and sCIs, it has been proposed that $SO_2$ may react with peroxy radicals ($RO_2$) (Kan et al., 1981). While the gas-phase reaction of $SO_2 + RO_2$ is usually too slow to compete with other $RO_2$ sinks (Berndt et al., 2015), Richards-Henderson et al. (2016) proposed that in polluted areas ($[SO_2] \geq 40$ ppb), $SO_2$ may react with $RO_2$ radicals at the surface of aerosol, and significantly accelerate (10-20 times higher) the heterogeneous oxidation rate of aerosol by OH radicals through chain propagation mechanism of alkoxy radicals. The reaction rate of particle-phase $SO_2 + RO_2$ ($\sim 10^{-13}$ cm$^3$ molecule$^{-1}$ s$^{-1}$) was calculated to be 4 orders of magnitude larger than that for the gas phase ($10^{-17}$ cm$^3$ molecule$^{-1}$ s$^{-1}$), indicating that this mechanism may be important for heterogeneous oxidation of aerosols, but the contribution to $SO_2$ sink is likely small.




Not only can $SO_2$ react with reactive species in the gas phase, it can also either partition into aqueous droplets and submicron particles or through heterogeneous reactions with the potential to alter SOA formation mechanisms and products. Reactions with dissolved $H_2O_2$ and $O_3$ are usually considered as the dominant sinks of $SO_2$ in aqueous droplets (Seinfeld and Pandis, 2006), and the reaction rates are a strong

function of particle acidity (Hung and Hoffmann, 2015; Seinfeld and Pandis, 2006). However, it was highlighted that in polluted areas, dissolved $NO_2$ and heterogeneous reactions on the surface of mineral dust could also contribute significantly to atmospheric $SO_2$ oxidation (He et al., 2014; Xue et al., 2016). More recently, Shang et al. (2016) proposed that without assistance of other oxidants, gaseous $SO_2$ can also be directly taken up by unsaturated fatty acid or long-chain alkenes through a [2+2] cycloaddition

mechanism under atmospheric conditions with observation of organic sulfur compounds as direct formation products.

Despite the importance of $SO_2$ in modulating BSOA formation, there have been few studies investigating the role of $SO_2$ in SOA formation from monoterpene ozonolysis, an important source of BSOA. In this

work, we study the direct interactions between $SO_2$ and reactive intermediates formed from ozonolysis of limonene and α-pinene, two important monoterpenes. We hypothesize that the presence of $SO_2$ changes SOA formation mechanisms and may lead to changes in SOA products and SOA yields. Interactions between $SO_2$ and reactive intermediates such as sCI and organic peroxides during SOA formation were investigated under different humidity conditions. We report synergistic effects between SOA formation

and $SO_2$ oxidation with observation of organosulfate formation. Results in this study provide a better mechanistic understanding of BSOA formation and atmospheric $SO_2$ oxidation.

## 2. Experimental Methods

Experiments were conducted both in a 1-m³ Teflon chamber for examining the time evolution of gaseous species and particles, and in a quartz flow tube for collection of particles onto filters and offline chemical

analysis.



### 2.1 Chamber experiments

Before each experiment, the chamber was flushed with purified air until the total particle number concentration, ozone concentration and $SO_2$ concentration was less than 10 # $cm^{-3}$, 1 ppb and 1 ppb, respectively. (R)-Limonene (97%, Sigma-Aldrich)/cyclohexane (99%, Caledon Laboratories Ltd.) or α-

pinene (99%, Sigma-Aldrich)/cyclohexane solution was injected into a glass vessel and then introduced into the chamber by purified compressed air at a flow rate of ~ 10 L $min^{-1}$. The injection ratio (v/v) of limonene/cyclohexane and α-pinene/cyclohexane was 1:1500 and 1:500, respectively. At these ratios, the reaction of OH with cyclohexane is calculated to be around 100 times faster than that of OH with monoterpene. $SO_2$ (5.2 ppm, balanced in $N_2$, Linde Canada) was injected into the chamber at 10 L $min^{-1}$

to achieve the desired initial concentrations. Ozone was added at a concentration more than 5 times higher than that of monoterpene to ensure complete consumption. Ammonium sulfate seed particles were introduced by a collison type atomizer (TSI 3076). In dry experiments (10-16% RH), seed particles were dried using a custom-made diffusion dryer before injection into the chamber. In humid experiments, seed particles were not dried when injected into the chamber. Chamber RH was controlled using a custom-

made humidifier and maintained at 47-55% which is above the efflorescence point of ammonium sulfate. Therefore, the liquid water content in seed particles in the humid experiments is expected to be higher than that in the dry experiments. However, it is noted that a diffusion dryer was placed before the particle sampling inlet to remove liquid water from the particles in order to eliminate its influence on calculating the change of organic particle volume/mass concentration. In all experiments, monoterpene concentration

was measured using a gas chromatograph with flame ionization detector (GC-FID, SRI 8610C) equipped with a Tenax TA trap sampled at a rate of 0.14 L $min^{-1}$ for 3 min. $SO_2$ and $O_3$ were measured by $SO_2$ analyzer (Model 43i, Thermo Scientific) and $O_3$ analyzer (Model 49i, Thermo Scientific), respectively. Particle size distribution and volume concentration were monitored using a custom-built scanning mobility particle sizer (SMPS) with a differential mobility analyzer (TSI 3081) and a condensation

particle counter (TSI 3772). Relative humidity and temperature were monitored using an RH/T transmitter (HX94C, Omega). The temperature was monitored to be 23 ± 2 °C. To maintain a positive pressure inside the chamber, a 1 L $min^{-1}$ dilution flow was added to balance the total sampling flow. Each experiment lasted for 4-5 h. Particle loss (including particle wall loss and dilution) in the chamber was corrected in a



size-dependent manner assuming first-order loss within each particle size bin, and the loss rate was

measured at the end of each experiment. Initial conditions and results are summarized in Table 1. It is
noted that no correction for semivolatile vapor wall loss was made in the chamber experiments in this
study. Therefore, the absolute values for SOA yields may be underestimated (Zhang et al., 2014).
However, with relatively high seed area concentrations (1535-3309 $\mu m^2\ cm^{-3}$) and volume concentrations
(45-122 $\mu m^3\ cm^{-3}$) used in this study, effects of vapor wall loss are expected to be similar across different

experiments and may not be important for relative SOA yield comparison. For example, similar SOA
yields were observed when the injected seed volume concentration ranged from 47 to 82 $\mu m^3\ cm^{-3}$ for
Exp. #1-3, and 59 to 71 $\mu m^3\ cm^{-3}$ for Exp. #11-13.

## 2.2 Flow tube experiments

To collect sufficient SOA mass for offline chemical analysis, SOA was also produced in a quartz flow

tube by reacting limonene or α-pinene with ozone (~3 ppm) in the presence or absence of $SO_2$ under dry
(10-13% RH) and humid (55-60% RH) conditions. The flow tube has a diameter of 10.2 cm and length
of 120 cm, and the residence time in the flow tube is 4 min. Two injection ratios of limonene and $SO_2$
(500 ppb/250 ppb and 500 ppb/100 ppb) were used to investigate the effects of $SO_2$ on SOA formation.
No seed aerosol was used during SOA formation to eliminate the influence of inorganic salt on chemical

analysis, particularly that of sulfate. SOA was collected onto pre-baked quartz filters for 24 h. Filters were
stored at -20℃ prior to analysis and extracted in 5 mL HPLC-grade methanol (>99.9%, Caledon
Laboratories Ltd.) by sonication for 10 min. The extract was filtered using a 0.2 μm pore size syringe
filter and prepared for composition analysis or peroxide quantification.

## 2.3 Chemical Characterization of SOA by ESI-IMS-TOF

Prior to chemical analysis, the SOA extract was concentrated to 0.5-1 mg mL$^{-1}$ under a gentle $N_2$ stream
in an evaporator (N-EVAP, Organomation). Particle composition was analyzed using electrospray
ionization-ion mobility spectrometry-high resolution time-of-flight mass spectrometry (ESI-IMS-ToF,
TOFWERK, hereafter referred to as IMS-TOF). Details of the IMS-TOF technique are described in recent
publications by Krechmer et al. (2016) and Zhang et al., (2016). Briefly, SOA solution was introduced


into IMS-TOF using direct infusion with a syringe pump (Legato 100, KDS) at 1-2 μL min$^{-1}$. Organic compounds in the SOA extract were ionized by ESI in the negative mode. Ion droplets were evaporated in a desolvation tube and then separated in the ion drift tube based on ion mobility. The ion mobility ($K$) of an organic compound is a function of its molecular structure and ion-neutral interactions with $N_2$ buffer gas, and is calculated by measuring the drift time ($t_d$) in the IMS drift tube:


$$K = \frac{1}{t_d} \frac{L_d^2}{V_d}$$

where $L_d$ is the length of the drift tube (20.5 cm) and $V_d$ is the drift voltage (-9600 V). In all analyses, the desolvation tube and the ion drift tube were maintained at $333 \pm 2$ K and atmospheric pressure (~1000

mbar). Generally, small and compact molecules have shorter ion drift times and higher ion mobilities than large and elongated molecules. One key feature of the IMS-TOF is that collision induced dissociation (CID) with nitrogen gas can be introduced between the ion drift tube and the time of flight region. CID analysis allows for attributing a fragment ion to its parent ion, as they share the same ion drift time (Zhang et al., 2016). Post-processing was performed with an Igor-based data analysis package (Tofware V2.5.7,

TOFWERK).

**2.4 Quantification of peroxides in SOA**

Peroxide content in SOA was quantified using an iodometric-spectrophotometric method adapted from Docherty et al. (2005). Briefly, the iodide ion ($I^-$) can be oxidized by a peroxide moiety (including $H_2O_2$, ROOH and ROOR) to form $I_2$ under acidic conditions. $I_2$ then complexes with $I^-$ to form $I_3^-$. $I_3^-$ is an

orange-brown color complex which absorbs strongly at 470 nm.

$$\text{Peroxide} \; + \; I^- \; \xrightarrow{\text{Acid}} \; I_2 \; \xrightarrow{I^-} \; I_3^-$$

Peroxide content was measured for limonene SOA formed under humid conditions. Limonene SOA

(LSOA) extract was concentrated to around 2 mg mL$^{-1}$ and added into a 96 well UV plate (160 μL/well,





No. 655801, Greiner Bio-One). 20 μL Formic acid (≥98%, Sigma-Aldrich) and 20 μL 0.1 g mL$^{-1}$ potassium iodide (KI) solution were added to initiate reaction. KI solution was prepared by dissolving KI (≥99%, Sigma-Aldrich) into MilliQ water (18.2 MΩ·cm). The plate was sealed with a UV transparent film (EdgeBio) to eliminate contact with ambient $O_2$. After 1 h at room temperature, the absorbance of

the solution was measured using an absorbance plate reader (SpectraMax 190, Molecular Devices) at 470 nm. The absorbance signal was calibrated using benzoyl peroxide, and converted to a mass fraction assuming a MW of 242.23 g mol$^{-1}$ (same as benzoyl peroxide). Background absorbance from a negative control (160 μL methanol + 20 μL formic acid + 20 μL KI) was subtracted from all reported absorbances. Each measurement was repeated at least two times.


### 2.5 Bulk solution $SO_2$ bubbling experiments

In addition to chamber and flow tube experiments, the reaction of $SO_2$ with peroxides was also investigated in bulk solutions. LSOA was collected from flow tube experiments mentioned previously and extracted using a methanol/$H_2O$ (1:1) solution. The solution was divided into two and added into

glass bottles. $SO_2$ (5.2 ppm balanced by $N_2$) was bubbled through one of the solutions at a flowrate of 0.02 L min$^{-1}$ for 2.5 h. $N_2$ was bubbled through the other solution in parallel for 2.5 h at the same flow rate as a negative control. The total peroxide content was measured using the iodometric-spectrophotometric method mentioned in previous section and compared between the two solutions. As a positive control, a solution of 2-Butanone peroxide (technical grade, Sigma-Aldrich) was also bubbled

with $SO_2$ and $N_2$ in parallel in the same manner.

### 2.6 Chamber experiments with $SO_3$

Experiments were also conducted to investigate the reactivity of $SO_3$ with organic compounds. The chamber was cleaned and filled with limonene (63 ppb) before experiment. Relative humidity in the

chamber was maintained at 10-12%, which is similar to the LSOA experiments under dry conditions in Table 1 (Exp. #4-10). $SO_3$ was generated by blowing fuming sulfuric acid (20% free $SO_3$ basis, Sigma-Aldrich) into the chamber. Briefly, 0.2 mL fuming sulfuric acid was injected into a glass vessel and then




blown with dry $N_2$ flow with a flow rate < 5 L min$^{-1}$. The upper limit of $SO_3$ injected into the chamber was estimated to be around 24 ppm.

**3. Results and Discussion**

**3.1 $SO_2$ decay and limonene SOA formation under dry conditions (RH < 16%)**

Reactions of $SO_2$ and SOA formation from limonene ozonolysis were investigated through experiments with and without $SO_2$. Synergistic effects were observed between LSOA formation and $SO_2$ oxidation, as $SO_2$ was consumed at the same time scales as the formation of LSOA. As shown in Fig. 1, as soon as

ozone was added into the chamber prefilled with limonene and $SO_2$ at $t = 0$, concentrations of limonene and $SO_2$ began to decrease simultaneously, and particle concentration began to increase, suggesting that limonene ozonolysis produces intermediates that react with $SO_2$. After about 100 min, limonene concentrations were below detection limits, and both $SO_2$ consumption and particle formation began to slow down. In order to confirm that limonene was needed to produce the intermediates reactive towards

$SO_2$, more limonene was added into the chamber at $t = 175$ min and both $SO_2$ consumption and particle formation resumed immediately. We therefore infer from the correlation between depletion rate of $SO_2$ and particle formation that similar species or processes are responsible for $SO_2$ reaction and LSOA formation.

When comparing SOA formation across different experiments under dry conditions (as shown in Fig. 2), we observed that aerosol formation increased with increasing initial $SO_2$ concentration when initial $[SO_2]$ < 140 ppb. At initial $[SO_2]$ > 140 ppb, it appears that $SO_2$ has no further effect on SOA yields with initial [Limonene] ~ 30 ppb. Here we expect that the observed enhancement in total aerosol formation by $SO_2$ is the result of formation and condensation of sulfuric acid and/or increased SOA formation owing to

increased particle acidity (Gao et al., 2004; Jang et al., 2002). Based on the measured loss of $SO_2$, we calculate the maximum contribution of condensed sulfuric acid to the increased aerosol mass by assuming all of the reacted $SO_2$ formed particle-phase sulfuric acid as a lower limit for SOA enhancement. As shown in Table 1, we compare two experiments (Exp. #1 and #7) in which similar amounts of limonene





was consumed in the presence of 139 ppb of $SO_2$ (Exp. #7) and in the absence of $SO_2$ (Exp. #1). In Exp.

#7, we observed a 5.7 ppb decay in $SO_2$ concentration, which would add a maximum of 14.7 $\mu m^3$ $cm^{-3}$ to

the particle volume concentration, assuming a density of 1.58 g $cm^{-3}$ for an aqueous sulfuric acid solution

under 10% RH (Heym, 1981). This amount of sulfuric acid can only account for 68% of the difference in

particle volume between Exp. #7 and #1. There is still 32% of the enhancement of particle volume

concentration that cannot be fully explained by introducing sulfate into the particle phase. Based on

previous work demonstrating that acid catalysis by sulfuric acid increases SOA yields, we expect that

SOA yields were enhanced as a result of increased particle acidity from condensation of sulfuric acid

(Czoschke et al., 2003; Surratt et al., 2007).

**3.2 $SO_2$ decay and limonene SOA formation under humid conditions (RH ~ 47-55%)**

$SO_2$ reaction and SOA formation were also examined under humid conditions. As shown in Table 1 and

Fig. 3A, an increase in particle volume concentration was also observed in the presence of $SO_2$ under

humid conditions. However, the enhancements of particle volume concentration were smaller compared

to those experiments conducted under dry conditions with similar initial conditions (e.g. Exp. #8 vs. Exp.

#14, and Exp. #10 vs. Exp. #15). One of the possible reasons is that the formation of high-MW organic

compounds and organosulfate is favored under high acidity conditions. The liquid water content in the

particle phase under humid conditions is suggested to be higher than that under dry conditions as

demonstrated in Section 2.1. Increased liquid water content reduces particle acidity through dilution and

leads to decreased SOA enhancement. In addition, in all humid experiments, a diffusion dryer was used

before the SMPS particle sampling inlet to remove water and eliminate the influence of condensed water

on particle volume concentration measurement. However, this may in turn lead to evaporation of

semivolatile species, resulting in the smaller changes in the particle volume concentration. The loss of

reactive intermediate onto the chamber walls may also play a role. We expect that the wall loss of reactive

intermediates may be higher under humid conditions than under dry conditions (Loza et al., 2010). It

should also be noted that in all the experiments, particle volume concentration instead of mass



concentration was measured. Particle density may increase as its composition changes, leading to apparent
       changes in SOA yields.

       On the other hand, greater $SO_2$ consumption was observed under humid conditions. For example, with an
       initial $SO_2$ concentration of 141 ppb, only 6 ppb of $SO_2$ was consumed under dry conditions (Exp. #8,
Table 1 and Fig. 3B). Meanwhile, under humid conditions, the decay in $SO_2$ concentration was 15 ppb
       with a similar set of initial conditions (Exp. #14, Table 1 and Fig. 3B). The difference in amounts of $SO_2$
       reacted and SOA yield enhancements between dry and humid conditions suggest that the mechanisms of
       the organic-$SO_2$ interactions are different between the two regimes. To identify these mechanisms, we
       conducted further experiments in which the experimental conditions were systematically varied to probe
specific mechanisms.

### 3.3 Mechanisms of $SO_2$ reaction

### 3.3.1 Under dry conditions: Interaction between $SO_2$ and Criegee intermediates

       Stabilized Criegee intermediates generated from alkene ozonolysis have been proposed to be important
       oxidants of $SO_2$ in the atmosphere (Mauldin III et al., 2012; Vereecken et al., 2012; Huang et al., 2015).
The rates of bimolecular sCI reactions depend strongly on molecular structure. The reaction of $CH_2OO$
       with water and water dimer is rapid, with rate constants of $<1.5 \times 10^{-15}$ cm$^3$ s$^{-1}$ and $6.5 \times 10^{-12}$ cm$^3$ s$^{-1}$
       (Chao et al., 2015), respectively, and is likely the dominant sink of $CH_2OO$ under almost all humidity
       conditions. However, Huang et al. demonstrated that $(CH_3)_2COO$, a di-substituted Criegee Intermediate
       has lower reaction rate constants with water and water dimer ($<1.5 \times 10^{-16}$ cm$^3$ s$^{-1}$ and $<1.3 \times 10^{-13}$ cm$^3$
s$^{-1}$), suggesting that the reaction of a di-substituted sCI with $SO_2$ can be competitive with the water
       reactions at atmospherically relevant RH. Since di-substituted sCIs can be produced from $O_3$ addition to
       either the endocyclic or exocyclic double bonds of limonene that yield sCI-1 and sCI-2, respectively
       (Scheme 1), we hypothesize that sCIs from limonene ozonolysis are responsible for the observed $SO_2$
       decay under dry conditions.




To examine the contribution of sCI + SO$_2$ to the observed SO$_2$ consumption, formic acid was added as a sCI scavenger for both dry and humid experiments (Exp. #18-20). The initial concentration of formic acid added in these experiments was ~13 ppm. Based on the previously measured rate constant for reaction between sCIs from monoterpenes and formic acid (about 3 times higher than that of sCIs + SO$_2$) (Sipilä

et al., 2014), we expect that at these formic acid concentrations, sCI + SO$_2$ reactions are minimized. As shown in Table 1 and Fig. S1 (see Supporting Information), much smaller SO$_2$ consumption was observed in the presence of 13 ppm of formic acid under dry conditions (Exp. #9 vs. Exp. #18, Table 1, Fig. S1A and S1B). This observation indicates without formic acid, the reaction with sCIs from monoterpene ozonolysis is a significant sink of SO$_2$. Results shown here are consistent with the observations from

Sipilä et al. who demonstrated the importance of sCI + SO$_2$ reactions through measuring the production of sulfuric acid (Sipilä et al., 2014). Therefore, in the presence of SO$_2$, limonene ozonolysis can produce sCIs that can directly oxidize SO$_2$ to sulfuric acid, which may then proceed to enhance aerosol acidity and SOA formation, as discussed in Section 3.1.

**3.3.2 Under humid conditions: Interaction between SO$_2$ and peroxides**

On the other hand, under more humid conditions (RH~50%), SO$_2$ consumption did not decrease even in the presence of a large excess of formic acid (Exp. #18 vs. Exp. #19, Table 1, Fig. S1B and S1C), suggesting that, unlike sCIs from smaller precursors (e.g. dimethyl substituted sCIs (Huang et al., 2015a)), sCIs from monoterpenes is not an important sink for SO$_2$ under humid conditions. The SO$_2$ consumption

remains significant and is even greater than under dry conditions, pointing to a yet unidentified sink of SO$_2$ that involves other reactive intermediates from monoterpene ozonolysis.

Here we propose that organic peroxides and/or hydrogen peroxide contribute significantly to the observed consumption of SO$_2$. To test this hypothesis, the total peroxide content in SOA produced in the presence

or absence of SO$_2$ was measured using the iodometric-spectrophotometric method mentioned previously (Section 2.4). Shown in Fig. 4, the mass fraction of total peroxides in LSOA is $(48 \pm 6)$ % in the absence of SO$_2$. When SO$_2$ is present during SOA formation (SO$_2$ : limonene = 250 ppb : 500 ppb), the peroxide





fraction decreases to $(13 \pm 1)$ %. To further confirm this interaction, bulk experiments were conducted by bubbling $SO_2$ into a solution of LSOA extract. Shown in Fig. S2 (left panel), the peroxide fraction

decreased significantly after $SO_2$ was bubbled through the LSOA solution, when compared to the negative control experiment using $N_2$ bubbling to account for potential evaporation and/or decomposition at room temperature. As a positive control, experiments were conducted by bubbling $SO_2$ through a solution of 2-butanone peroxide. Again, a significant decrease in the peroxide content was observed (Fig. S2, right panel), confirming that organic peroxides are reactive towards $SO_2$.


Since the observed $SO_2$ decay is greater under humid conditions than dry conditions during the chamber experiments and higher liquid water content is expected under humid conditions, it is likely that $SO_2$ first dissolves into the aqueous particle and the reaction proceeds in the aqueous phase. It is well known that the aqueous phase reaction of hydrogen peroxide is the dominant sink of $SO_2$ in the atmosphere (Seinfeld

and Pandis, 2006). However, to the best of the authors' knowledge, this work is the first experimental chamber study to suggest organic peroxides from monoterpene ozonolysis in aqueous particles are reactive towards $SO_2$ under atmospherically relevant RH conditions. With the iodometric-spectrophotometric method used in this study, we cannot distinguish between different types of peroxides, specifically ROOH and ROOR. Previous work has shown that ROOH are important products of

monoterpene ozonolysis and precursors to peroxyhemiacetal formation (Docherty et al., 2005; Tobias et al., 2000) and a major component of SOA formed from low NOx photooxidation of many VOCs, including isoprene (Surratt et al., 2006) and n-alkanes (Schilling Fahnestock et al., 2014). SOA from reactions between isoprene and nitrate radicals has been shown to contain significant amounts of ROOR-type peroxides (Ng et al., 2008). Further work should focus on the mechanisms and kinetics of reaction

between $SO_2$ and different types of organic peroxides, which are ubiquitous in the atmosphere.

### 3.3.3 Other mechanisms of $SO_2$ reactions: $SO_2$ + ozone, $SO_2$ + OH

Experiments were also conducted to rule out other possible explanations for $SO_2$ decay. Aqueous phase $SO_2$ has been shown to react with dissolved ozone at appreciable rates at pH $\geq$ 5 (Seinfeld and Pandis,



2006). To rule out the reaction between $SO_2$ + ozone, formic acid (13 ppm to minimize sCI reactions), ammonium sulfate seed (246 $\mu m^3$ $cm^{-3}$, $4.9 \times 10^4$ # $cm^{-3}$), $SO_2$ (289 ppb) and ozone (485 ppb) were injected and kept in the chamber under humid conditions (50% RH) for 6 h. Over the course of the experiment, the change in [$SO_2$] was less than 1 ppb, which is within experimental uncertainty (Fig. S3A), suggesting that reactions between $SO_2$ and ozone either in the gas or particle phase have negligible effects

on $SO_2$ consumption. Another potential sink of $SO_2$ is the gas-phase reaction with OH radicals, which may be produced from unimolecular decomposition of the Criegee Intermediate. We conducted experiments with an initial concentration of 30 ppb limonene, 68 ppm cyclohexane and 300 ppb $SO_2$. At these concentrations, reaction rate of cyclohexane and OH is calculated to be around 130 times higher than that of $SO_2$ and OH with $\boldsymbol{k_{cyclohexane+OH}}$ = $6.97 \times 10^{-12}$ $cm^3$ $molecule^{-1}$ $s^{-1}$ (Atkinson and Arey,

2003) and $\boldsymbol{k_{SO_2+OH}}$ = $1.2 \times 10^{-12}$ $cm^3$ $molecule^{-1}$ $s^{-1}$ (Atkinson et al., 2004). Therefore, under our experimental conditions, the role of OH reaction is minimized. To confirm that OH reactions are not important, the concentration of cyclohexane, the OH scavenger, was doubled in additional limonene ozonolysis experiments (Exp. #7-8 vs. Exp. #16-17). No decrease of $SO_2$ consumption was observed ((14 $\pm$ 2) % in Exp. #7-8 vs. (15 $\pm$ 1) % in Exp. #16-17), confirming that gas phase OH radicals play a minor

role in $SO_2$ oxidation in this study.

### 3.4 Organosulfate formation

Reactions between organic and sulfur-containing compounds are illustrated by the observed formation of organosulfates in the presence of $SO_2$. Previous work has shown that the bisulfate ion can react with alcohol or epoxide to form organosulfates, and organosulfate formation may be enhanced by particle

acidity (Surratt et al., 2008). In this work, organosulfates present in SOA were identified using ESI-IMS-TOF through elemental formulas that are calculated from high resolution $m/z$ ratios, and by matching ion drift times to either of the two most abundant sulfate fragments ($HSO_4^-$ and $CH_3SO_4^-$) in the mass spectra.

Shown in Fig. 5A and 5B, eight parent ion peaks were matched to $HSO_4^-$ and $CH_3SO_4^-$ fragment ion peaks

in LSOA. The mass-to-charge ratios of these ions are consistent with sulfate-containing elemental formulas, confirming the organosulfate moiety. In addition, the assigned organosulfate ions were further

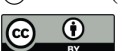



validated by identifying trends (C, $CH_2$, O, $CH_2O$ and $CO_2$) in Kendrick mass defect plots (Walser et al., 2008), as shown in Fig. S4 and Table S1. For example, in Fig. 5C, the ion with $m/z$ 297.0835 was observed to have the same IMS drift time as $CH_3SO_4^-$ (drift time = 30.14 ms), indicating that $m/z$ 297.0835 is an

organosulfate ion. The $m/z$ ratio is also consistent with the molecular formula of $C_{10}H_{17}O_8S^-$, and falls within the trend lines of adding $CH_2O$ and O groups in the Kendrick mass defect plots to other identified organosulfate parent ions (e.g. $C_9H_{15}O_7S^-$ and $C_{10}H_{17}O_7S^-$). These ions were present only when $SO_2$ was added during SOA formation.

It should be noted that the number of organosulfate ions identified increased with increasing $SO_2$ concentrations. Shown in Fig. S5, we were able to identify 8 organosulfate ions when LSOA was formed in the presence of 100 ppb of $SO_2$, and 16 organosulfate ions when $SO_2$ was 250 ppb. We also observed that the total signal fraction of these organosulfate ions increased, but since no authentic standards were available for quantification, no conclusions can be drawn about the difference in organosulfate amounts

between the two experiments. By comparing ESI mass spectra, we observe that when $SO_2$ is present, there is a significant decrease in signal fraction from the high-MW species ($m/z$ 320-500) and an increase in the signal fraction from low-MW compounds ($m/z$ 150-320). The change of MW distribution may be due to the formation of organosulfate, and/or the formation and/or the uptake of low-MW compounds. As shown in Fig. S4, all the identified organosulfates are within the mass range of m/z 150-320. Although

the hygroscopicity of organosulfate is not known, the sulfuric acid produced in the experiments with $SO_2$ may take up water and encourage uptake of small water-soluble organics, such as peroxides, epoxides and small aldehydes, also leading to the change of MW distribution in the ESI mass spectrum. Formation of low-MW compounds is also possible in the experiments with $SO_2$. For example, peroxide may react with bisulfite in the particle phase instead of forming peroxyhemiacetals, that will affect MW distribution.

In all experiments, the average carbon oxidation state (OSc = 2 O/C - H/C) of SOA was observed to increase with $SO_2$. It is noted that since the negative mode in ESI is sensitive only to acidic species, the effects of $SO_2$ on relative signal fractions and oxidation states observed here may only be valid for these species.



### 3.6 Potential mechanisms of SOA yield enhancement: comparison to α-Pinene

As mentioned earlier, enhanced SOA formation was observed from limonene ozonolysis in the presence of $SO_2$. The relative signal fraction of high-MW products measured in the IMS-TOF was reduced compared to when $SO_2$ was not present, suggesting that $SO_2$ may reduce oligomer formation, which may decrease SOA yields. However, we also observe that the presence of $SO_2$ (which is then converted to sulfate via previously mentioned mechanisms) increases organosulfate formation, and the average carbon oxidation state of low-MW products also increases. Our results therefore suggest that for limonene ozonolysis, the effect of functionalization (formation of organosulfate and increase in oxidation state) exceeds that of decreased oligomerization, leading to an overall increase in SOA yields. It should be noted that previous studies have focused mostly on the effect of acidic sulfate on SOA yields, which likely promotes both functionalization and oligomerization reactions. Here we show that while $SO_2$ leads to a decrease in oligomerization, but there is still an overall increase in SOA yields from limonene ozonolysis.

To further compare the effects of oligomerization and functionalization, SOA formation from α-pinene ozonolysis was examined in the presence of $SO_2$. The IMS-TOF mass spectra of α-pinene SOA (ApSOA) (Fig. S6) show a similar decrease in the high *m/z* signal fraction when $SO_2$ is present, suggesting that $SO_2$ has a similar effect of decreasing oligomerization in this system. However, unlike in limonene ozonolysis, α-pinene SOA yields did not change significantly under different $SO_2$ concentrations under both dry and humid conditions (Fig. 7 and Table 1). It is likely that any enhancement in SOA yield by $SO_2$ through functionalization is masked by reduced oligomerization. As a result, there is little overall change in SOA yields from α-pinene ozonolysis. It is likely that the difference between the two systems can be explained by the number of double bonds and the extent of functionalization. Limonene has two double bonds. If $SO_2$ prevents oligomerization of the first-generation products, these products can still react further with ozone to add another oxidized functional groups to form condensable products. On the other hand, first-generation oxidation products from α-pinene ozonolysis may be too volatile to condense, and the presence of $SO_2$ reduces oligomerization and prevents any enhancements in SOA yields.



Under dry conditions, $SO_2$ can be oxidized by sCI to $SO_3$, which may be reactive towards organic compounds and may change the formation mechanism of SOA. As shown in Fig. S3B, $SO_3$ reacted rapidly with $H_2O$ to form sulfuric acid once injected, even at 11% RH, the lowest RH among all the experiments

conducted. During the experiment, around 7 ppb (11% of initial) limonene was reacted, which can be attributed to the reactive uptake of limonene onto sulfuric acid seed. This is consistent with the experimental observation from Liggio and Li (2008) in which significant uptake of monoterpenes onto highly acidic seed was observed in chamber studies under various humidity conditions. It is noted that the $SO_3$ concentration was very high (estimated to be ~24 ppm) during this experiment, which can be

inferred from the rapid formation of new particles ($2.8 \times 10^6$ # $cm^{-3}$ and volume concentration of $1.1 \times 10^4$ $\mu m^3$ $cm^{-3}$), which are likely nucleated sulfuric acid particles. The concentration of $SO_3$ used in this test (~24 ppm) was expected to be much higher than those generated in the experiments shown in Table 1 (an estimated upper limit of 60 ppb $SO_3$ formation with initial limonene concentration of 30 ppb, assuming that sCI yield is unity and sCI only reacts with $SO_2$). Therefore, it is likely that the reaction

between $SO_3$ and organic compounds do not play an important role in SOA formation under the experimental conditions in this study.

**4. Implications**

Our combined experimental observations of $SO_2$ consumption and formation of LSOA and ApSOA suggest that both sCI and organic peroxides formed from monoterpene ozonolysis may play crucial roles

in $SO_2$ oxidation under atmospherically relevant humidity levels. We propose the simplified mechanisms shown in Scheme 2 to summarize our findings. Under dry conditions, sCI reacted with $SO_2$ to form $SO_3$ which quickly reacts with water to form sulfuric acid. In the presence of sulfuric acid in the particle phase, SOA formation can be enhanced through acid-catalyzed reactions (Jang et al., 2002). At the same time, we observed reduced oligomerization for semivolatile oxidation products relative to increased low-MW

compounds by $SO_2$. As more $SO_2$ was added, the formation of sulfuric acid was limited by the initial monoterpene concentration, resulting in little change in particle acidity and, consequently, SOA yields. On the other hand, under atmospherically relevant humidity conditions, most of the sCI is scavenged by water and/or water dimer, and the sCI + $SO_2$ reaction is likely insignificant. However, $SO_2$ can partition



into aerosol liquid water to form $HSO_3^-$, and we present evidence to suggest that $HSO_3^-$ can further react

with organic peroxides produced from monoterpene ozonolysis. This mechanism is consistent with the

greater $SO_2$ consumption observed under humid conditions, since more aerosol water was available for

both $SO_2$ and peroxides to partition.

In order to evaluate the relative contributions of sCI and peroxides as reactive sinks for $SO_2$ in our

experiments, we formulate a simplified kinetic model to attribute observed $SO_2$ loss to each process. We

note that without detailed knowledge of the $SO_2$ uptake mechanisms, the heterogeneous reaction of $SO_2$

with condensed phase peroxide is simplified as a bimolecular reaction. This reaction may depend on many

factors, such as aerosol pH, aerosol liquid water content, and ionic strength. Nonetheless, we use this

simplified model to apportion the observed $SO_2$ loss under the experimental conditions employed in this

work to each process. In particular, we will use this model to illustrate the relative importance of the sCI

reaction under the two different experimental RH. Results are shown in Fig. 8 and the details of the box

model can be found in the supporting information (Section S6).

In this model, we calculated the relative contributions of the two pathways under two humidities (10%

and 50%) and under two initial concentrations of $SO_2$. As our laboratory observation suggests, $SO_2$

consumption increases with increasing humidity (Fig. 8A and 8B). In the high $SO_2$ scenario (Fig. 8A,

$[SO_2] > $ [limonene]), both interactions with sCI and peroxide play important roles in $SO_2$ oxidation. A

major fraction of $SO_2$ consumption can be attributed to the reaction with sCI under dry conditions. At

50% RH, the amount of $SO_2$ consumed by the sCI pathway drops slightly (from 3 to 2 ppb in this

scenario). On the other hand, the relative importance of reactive uptake by peroxides become dominant

at 50% RH, accounting for 87% of the total $SO_2$ consumption. In the low $SO_2$ emission scenario (Fig. 8B,

$[SO_2] < $ [limonene]), sCI does not react with $SO_2$ at appreciable amounts, owing to the competition from

reactions with water and water dimer. Therefore, sCI chemistry does not contribute significantly to the

$SO_2$ sink, and $SO_2$ consumption is dominated by reactions with peroxides and other reactive

intermediates. To identify when the transition from "low $SO_2$" to "high $SO_2$" occurs, simulations were

performed for a range of $SO_2$ concentrations, shown in Fig. 8C. Based on these results, we identified that

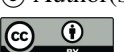



even at RH = 10%, sCI does not become an important sink of $SO_2$ until $SO_2$ exceeds 50 ppb (with 30 ppb limonene injection), and this threshold is likely greater than 500 ppb at RH = 50%. The reaction with peroxides is modelled as a simplified bimolecular reaction to match the observed $SO_2$ decay in our
experiments. Moving forward, more information about the reaction mechanism is needed to accurately model this reaction. In particular, the specific peroxide compounds that are reactive towards $SO_2$ need to be identified using advanced analytical techniques (Krapf et al., 2017; Reinnig et al., 2009). Also, since it is likely that the reaction is occurring in aqueous particles, the Henry's Law constant of the peroxide compounds will need to be measured. Despite these missing parameters, our simplified model highlights
the importance of the reactive uptake pathway, and suggests further studies are warranted to elucidate the reaction rates and mechanisms for this reaction.

Currently, as a result of air quality control policies, $SO_2$ concentrations have significantly decreased in many areas in the world during the past decades. For example, the annual national average $SO_2$
concentration has dropped to < 10 ppb in the U.S. (U.S. EPA, 2013) and 1.3 ppb in Canada (ECCC, 2016). However, high $SO_2$ concentrations can still be observed, especially in some hot spots in North America such as near oil sands operations in Northern Alberta (Hazewinkel et al., 2008), and in developing countries like China where coal combustion is the main energy source. Hourly $SO_2$ concentrations frequently exceed 100 ppb in some megacities in China during the winter season (Lin et al., 2011, Zhang
et al., 2015). Recent studies have shown that during heavy haze episodes, the rapid oxidation of $SO_2$ to sulfate cannot be explained by known mechanisms (Guo et al., 2014; Wang et al., 2014), and while heterogeneous reaction mechanisms have been proposed (Wang et al., 2016) , these mechanisms require relatively high pH to be plausible. Based on our experimental observations of $SO_2$ decay, we estimate that the uptake coefficient of $SO_2$ on the aqueous particle through reacting with peroxides from limonene
ozonolysis is $1\text{-}5 \times 10^{-5}$ (Supporting Information, Section S7). These values are comparable to those from heterogeneous uptake of $SO_2$ on mineral aerosol (Huang et al., 2015b, 2014; Adams et al., 2005; Ullerstam et al., 2003) and sea salt (Gebel et al., 2000). It should be noted that organic peroxides are ubiquitous in different SOA systems and can be formed from oxidation by OH (Surratt et al., 2006; Yee et al., 2012), $O_3$ (Docherty et al., 2005) and $NO_3$ (Ng et al., 2008). Results from our study therefore suggest a new



pathway of SO$_2$ oxidation in the atmosphere, which may contribute to the missing mechanisms of high-sulfate production in the polluted areas. Future work should investigate the role of peroxides from different SOA systems in oxidizing SO$_2$ and the atmospheric importance of these reactions.

The importance of the reaction pathways (sCI and reactive uptake) proposed in this study imply that
oxidation of VOCs and reactions of SO$_2$ are tightly coupled. It is important to note that SO$_2$, the precursor to sulfate, can directly influence the chemistry of SOA formation. And the oxidation of monoterpenes provides viable pathways to act as SO$_2$ sinks and a source for sulfate in the atmosphere. Therefore, oxidation of VOCs and SO$_2$ must be considered holistically in order to fully understand the impacts of anthropogenic emissions on atmospheric chemistry.


**Acknowledgement**

This work was funded by Natural Sciences and Engineering Research Council and Canadian Foundation for Innovation. JY would like to acknowledge financial support from the Ontario Trillium Scholarship. The authors would like to thank Dr. Barbara Turpin for insightful comments on heterogeneous uptake of
SO$_2$ onto aqueous particles and thank Dr. John Liggio for helpful discussion with the SO$_3$ experiments.

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



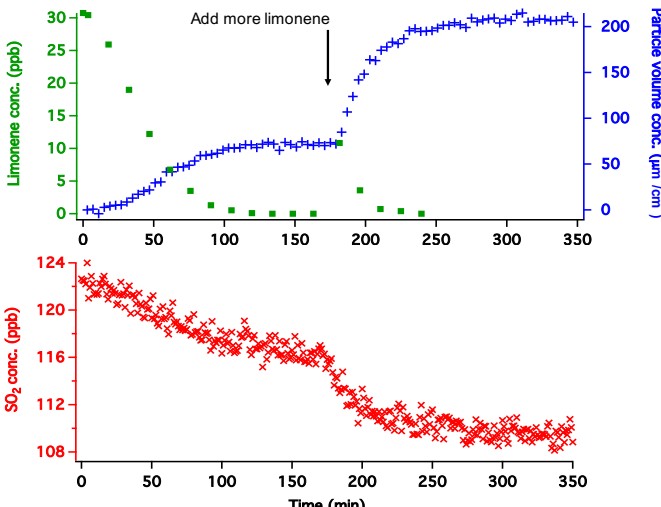


**Figure 1** Particle volume concentration, limonene concentration and SO$_2$ concentration as a function of experimental time. After 105 min, the limonene concentration was below 1 ppb, and the particle volume concentration and SO$_2$ concentration began to stabilize. At $t$ = 170 min, additional limonene was injected into the chamber, and particle formation and SO$_2$ decay resumed. The timescale of SO$_2$ consumption matches that of SOA formation suggesting that there are synergistic effects
between LSOA formation and SO$_2$ oxidation.

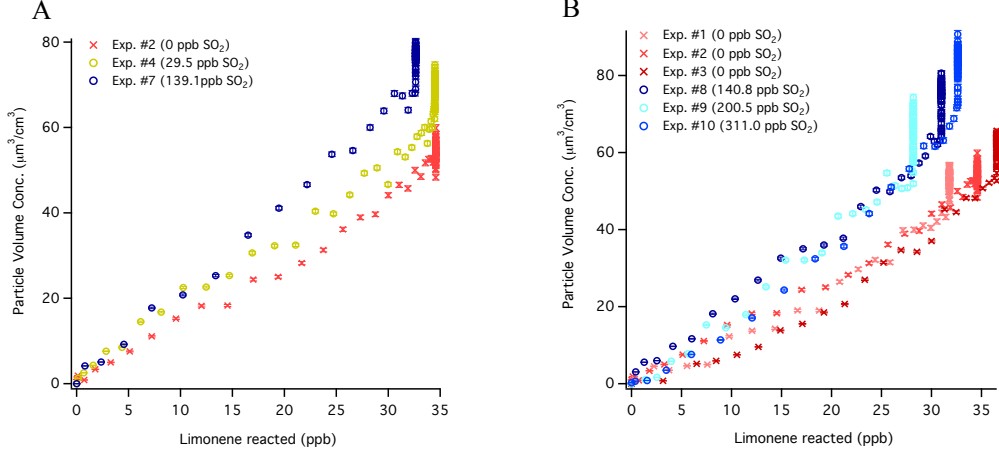

**Figure 2** Growth of particle volume concentration as a function of limonene consumption under dry conditions. Particle concentration increased with increasing SO$_2$ injection concentration (0 ppb, 30 ppb and 139 ppb) under dry conditions (panel A). The enhancement reached a plateau as more SO$_2$ was injected (panel B).



**Table 1** Summary of conditions and results for limonene and α-pinene SOA experiments

| Exp. # | HC reacted (ppb) | Initial SO$_2$ conc. (ppb) | Δ SO$_2$ (ppb) | HCOOH (ppm) | Seed volume conc. [a] (μm$^3$ cm$^{-3}$) | Final volume conc. [a] (μm$^3$ cm$^{-3}$) | ΔM [b] (μg cm$^{-3}$) | SOA yield | RH |
|---|---|---|---|---|---|---|---|---|---|
| **Limonene** | | | | | | | | | |
| 1 | 31.8 | 0 | 0 | 0 | 67.0 | 121.9 | 71.4 | 39.6% | 14% |
| 2 | 34.6 | 0 | 0 | 0 | 82.3 | 137.3 | 71.5 | 36.5% | 13% |
| 3 | 36.5 | 0 | 0 | 0 | 46.9 | 110.7 | 82.9 | 40.1% | 16% |
| 4 | 34.5 | 29.5 | 2.7 | 0 | 59.8 | 132.1 | 94.0 | 48.1% | 16% |
| 5 | 31.1 | 110.3 | 5.2 | 0 | 46.8 | 116.9 | 91.1 | 51.7% | 11% |
| 6 | 30.4 | 122.2 | 5.9 | 0 | 65.5 | 136.8 | 92.7 | 53.8% | 10% |
| 7 | 31.0 | 139.1 | 5.7 | 0 | 89.8 | 166.3 | 99.5 | 56.6% | 12% |
| 8 | 32.6 | 140.8 | 6.1 | 0 | 42.9 | 119.9 | 100.1 | 54.2% | 15% |
| 9 | 28.2 | 200.5 | 5.5 | 0 | 122.6 | 193.1 | 91.7 | 57.4% | 13% |
| 10 | 32.6 | 311.0 | 7.3 | 0 | 68.2 | 151.5 | 108.3 | 58.6% | 16% |
| 11 | 31.5 | 0 | 0 | 0 | 68.1 | 115.1 | 61.1 | 34.2% | 55% |
| 12 | 37.3 | 0 | 0 | 0 | 59.0 | 118.3 | 77.0 | 36.4% | 50% |
| 13 | 38.4 | 0 | 0 | 0 | 70.9 | 125.8 | 71.4 | 32.8% | 50% |
| 14 | 33.7 | 144.3 | 15.5 | 0 | 57.4 | 110.5 | 69.0 | 36.1% | 55% |
| 15 | 27.8 | 308.8 | 15.2 | 0 | 78.5 | 130.0 | 67.0 | 42.5% | 47% |
| 16 | 35.0 | 137.4 | 7.4 [c] | 0 | 56.9 | | | | 16% |
| 17 | 29.1 | 136.4 | 6.5 [c] | 0 | 57.7 | | | | 14% |
| 18 | 34.7 | 251.6 | 2.2 | 13 | 65.0 | | | | 12% |
| 19 | 25.4 | 262.2 | 10.0 | 13 | 52.3 | | | | 50% |
| 20 | 28.3 | 605.4 | 12.4 | 13 | 117.5 | | | | 52% |
| **α-Pinene** | | | | | | | | | |
| 21 | 23.9 | 0 | 0 | 0 | 30.0 | 53.0 | 28.8 | 21.3% | 14% |
| 22 | 25.0 | 0 | 0 | 0 | 41.0 | 65.4 | 30.5 | 21.5% | 14% |
| 23 | 29.6 | 0 | 0 | 0 | 24.0 | 50.5 | 33.1 | 19.7% | 12% |
| 24 | 27.2 | 24.5 | 1.2 | 0 | 35.0 | 55.9 | 26.1 | 16.9% | 13% |
| 25 | 24.1 | 42.3 | 1.9 | 0 | 64.6 | 81.1 | 20.6 | 15.1% | 10% |
| 26 | 21.8 | 107.8 | 2.3 | 0 | 54.3 | 70.7 | 20.5 | 16.6% | 12% |
| 27 | 33.7 | 99.2 | 2.6 | 0 | 41.0 | 71.7 | 38.4 | 20.1% | 12% |
| 28 | 27.4 | 0 | 0 | 0 | 55.2 | 76.8 | 27.0 | 17.4% | 49% |
| 29 | 27.7 | 0 | 0 | 0 | 38.6 | 63.2 | 30.8 | 19.6% | 48% |
| 30 | 27.2 | 53.1 | 4.5 | 0 | 50.1 | 71.6 | 26.9 | 17.4% | 46% |

[a]: Volume concentrations after particle wall loss correction.

[b]: Particle mass concentration formed during the experiments (ΔM = (Final volume conc. - Seed volume conc.) × ρ). A density of 1.30 and 1.25 g cm$^{-3}$ was applied to calculate limonene and α-pinene SOA mass concentration, respectively.

[c]: Twice the amount of cyclohexane was added in Exp. #16 and #17 to examine the reaction between SO$_2$ and OH.






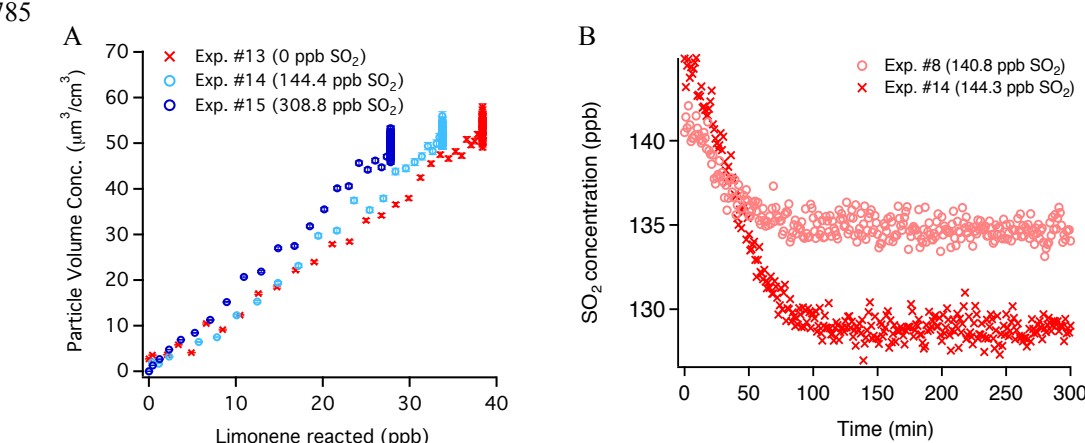

**Figure 3** A) Particle volume concentration as a function of limonene reacted under humid conditions. Enhanced particle mass formation was observed with increasing $SO_2$ concentration. B) Greater consumption of $SO_2$ was observed under humid
conditions than under dry conditions. By comparing Exp. #14 (humid) and Exp. #8 (dry), with similar initial $SO_2$ concentration, $SO_2$ consumption was more than two times greater under humid conditions.


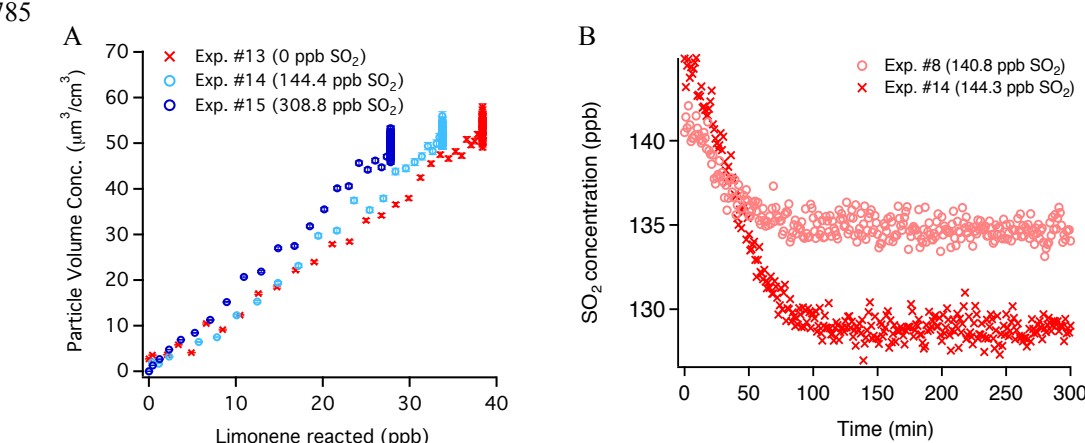

**Scheme 1** Formation of di-substituted sCI from limonene ozonolysis




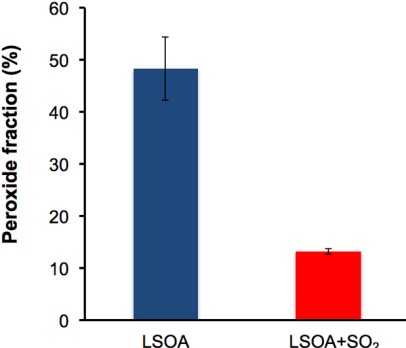

**Figure 4** Mass fraction of peroxides in LSOA formed in the presence and absence of $SO_2$. Error bars represent measurements of three LSOA and three LSOA + $SO_2$ filters collected from different experiments. Peroxide measurement for each filter was repeated two times.

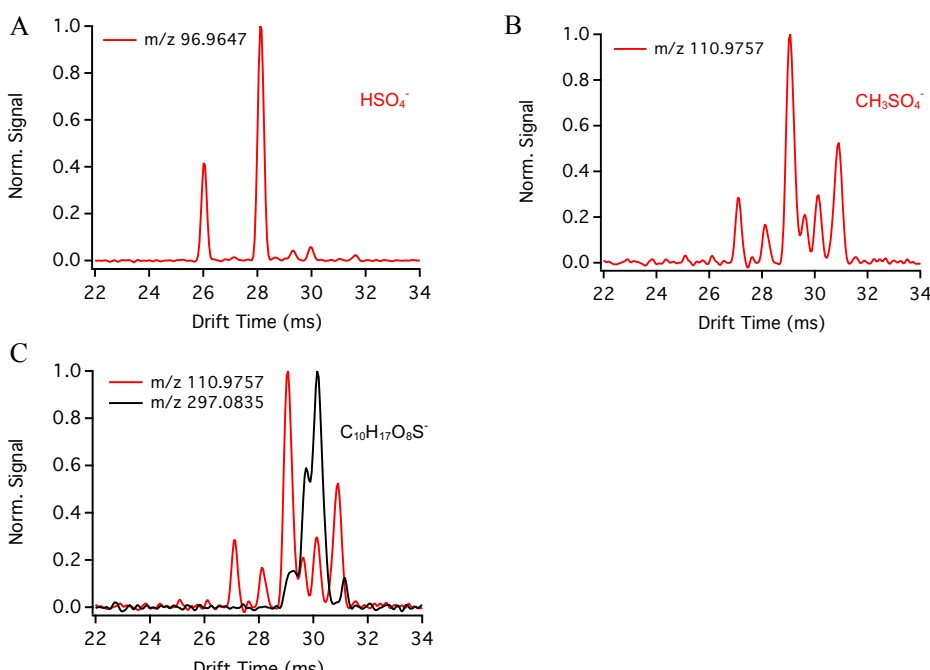

**Figure 5** Organosulfate identification using IMS-TOF: A) Drift time of $HSO_4^-$ (m/z 96.9647); B) Drift time of $CH_3SO_4^-$ (m/z 110.9757); C) Drift time of $CH_3SO_4^-$ and m/z 297.0835 (assigned to $C_{10}H_{17}O_8S^-$). Overlapping of the drift times at 30.14 ms between the two mass-to-charge ratios indicates that the daughter ion $CH_3SO_4^-$ is a fragment ion of the parent ion $C_{10}H_{17}O_8S^-$.




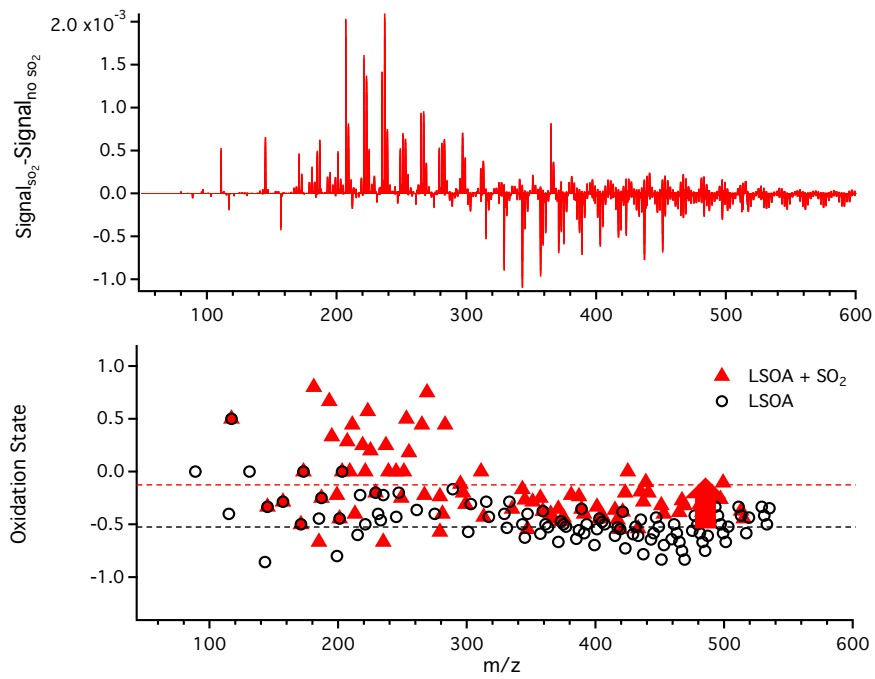

**Figure 6** Difference of normalized mass spectra between LSOA in the presence and absence of SO$_2$ (top panel). Signal of
HSO$_4^-$ (m/z 96.96) was excluded such that only organic mass spectra are compared. Bottom panel shows the average carbon
oxidation state of each peak detected in IMS-TOF and the overall average carbon oxidation states of LSOA (black dashed line)
and LSOA + SO$_2$ (red dashed line).







Figure 7 Growth of particle volume concentration as a function of reacted α-pinene under dry conditions. No significant change of SOA formation was observed.


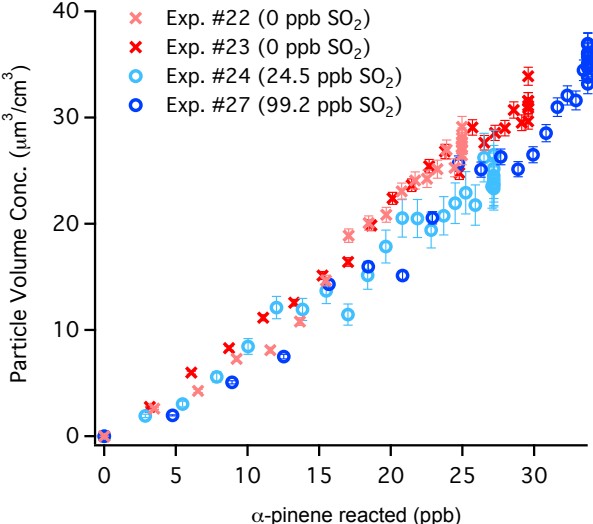

Scheme 2 The proposed reaction mechanisms for $SO_2$ and reactive intermediates in monoterpene ozonolysis.




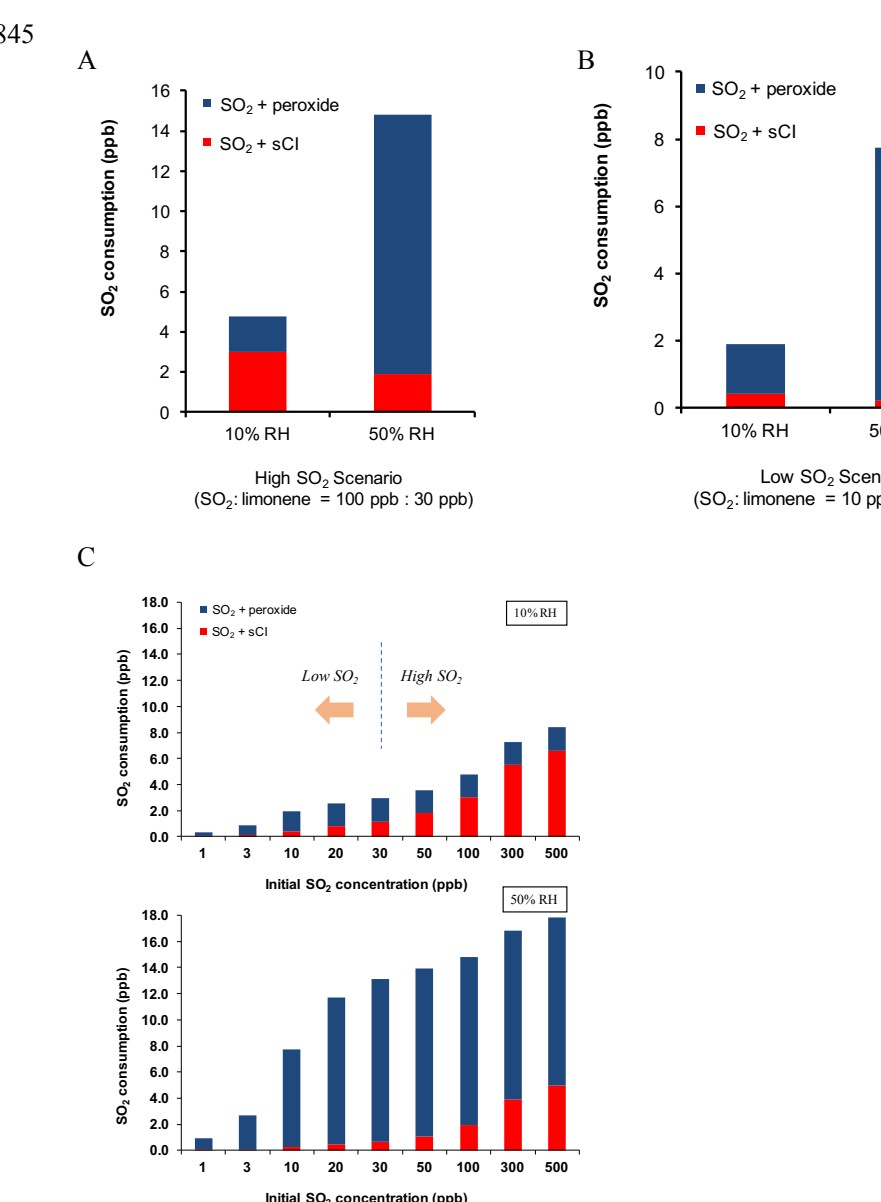


**Figure 8** SO$_2$ consumption through reactions with sCI and peroxide under different humidity conditions in a high SO$_2$ scenario (100 ppb, panel A) and a low SO$_2$ scenario (10 ppb, panel B). And SO$_2$ consumption as a function of initial SO$_2$ concentration under different humidity conditions (panel C).