# Peer review of "Novel Pathway of SO2 Oxidation in the Atmosphere: Reactions with Monoterpene Ozonolysis Intermediates and Secondary Organic Aerosol"

_Atmospheric Chemistry and Physics, 2017_

## Referee Comment (RC1) · Anonymous Referee #1 · 21 Dec 2017

This is a very nice investigation of the interactions between SO2 and reactive intermediates from monoterpene ozonolysis under different humidity conditions (10% vs. 50%). Increasing humidity seems to change the chemical regime in the system, shifting from a stabilized Criegee Intermediates (sCI) produced from ozonolysis, to peroxybased particle phase chemistry leading to organosulfates production. I really enjoyed reading this contribution, which is nicely and convincingly presented. Therefore I do recommend its publication subject to very minor changes.

Experiments were made in presence of excess ozone to ensure the complete consumption of the terpenes. Have you performed tests experiments where this was not

the case? Would expect any difference in the chemical regimes to SO2 oxidation at lower ozone concentration? For instance due to reduced peroxy-functions productions, concomitant presence of unsaturations, etc. . .

Maybe you could add in the experimental section 2.2, the ozone levels used in the flow tube experiments with, maybe, some indications how this compares to the chamber ones?

You are making clear that limonene is needed to induce a SO2 loss (line 225), but does this fully exclude that loss of SO2 is not firstly physically driven by solubilization into the nascent aerosols? (Adding limonene would also affect such an equilibrium due to the growth rate of the particles).

The paragraph starting at line 230 is slightly confusing to me. How do you conclude/affect to the growth effect to sulfuric acid? How would that acid be produced efficiently in your system? OH reaction can be excluded due to the presence of an OH scavenger and ozone reacts quite slowly under acidic conditions. Would all this just be linked to sCI chemistry? Maybe adding a few words of explanation would be helpful.

Humidity seems to reduce the SOA enhancement, as measured in the chamber. But what about the products distribution is humidity reducing the amount of organosulfate being produced? Also as noted, an increased humidity leads to an increased particle phase pH and hence an increased reactivity of ozone toward SO2 leading to sulfate production that may explain partly the observed enhanced SO2 loss under humid conditions. Would that be a sign of a competition between ozone reactivity and peroxy-type chemistry leading to organosulfate production, as this exist for cloud processing of SO2 (which is strongly pH dependent)?

Line 76 : please add the reference for the statement : "The reaction rate of particle-phase SO2 + RO2" Line 88: add also the reference to: Passananti, M.; Kong, L. D.; Shang, J.; Dupart, Y.; Perrier, S.; Chen, J. M.; Donaldson, D. J.; George, C. Organosulfate Formation through the Heterogeneous Reaction of Sulfur Dioxide with Unsaturated

Fatty Acids and Long-Chain Alkenes. Angewandte Chemie-International Edition 2016, 55 (35), 10336-10339.

---

## Referee Comment (RC2) · Anonymous Referee #2 · 27 Dec 2017

The paper proposes a mechanism of $SO_2$ oxidation by organic peroxides in aerosol particles, through which the formation of biogenic secondary organic aerosol is enhanced. Reactions between $SO_2$ and stabilized Criegee Intermediates or organic peroxides are suggested to be the dominant sinks for $SO_2$ under dry or humid conditions. Although the oxidation of $SO_2$ has been addressed by a number of studies over the past decades, rapid oxidation of $SO_2$, readily observed in heavily polluted area, remains unexplained by existing mechanisms. The current contribution is a welcome addition to the field.

The authors suggested that under dry conditions, particle acidity is increased due to the condensation of $H_2SO_4$ to the aerosols particles, thus enhancing SOA formation through acid catalyzed reactions. Under dry conditions (~ 10% RH), while H2SO4 is hygroscopic, the particles will contain less liquid water, resulting in possibly low dissociation of $H_2SO_4$ and hence low hydrogen activity. And in the humid condition (~ 50% RH), the authors explained that due to high liquid water content of the particles, the acidity of the aerosols were decreased thus resulting in decreased SOA yields. The aerosol water can potentially increase the dissolution of $H_2SO_4$, thus increasing the acidity too. However, as organics are formed, they may be hygroscopic and their presence reduces the DRH of AS so the solution will be less acidic. Overall, role of condensation of $H_2SO_4$ and effects of SOA on changes of phase and aerosol liquid water of the particles need to be incorporated in the discussion.

Overall:

1.      OS vs. inorganic sulfate. OS and oligomers analyses are not quantitative, it is not easy to rely on their peak identifications to explain the difference in yields under wet and dry conditions.

2.      Aerosol acidity has been used heavily to explain the results. Need better, at least, semi-quantitative discussions, on aerosol acidity. Dissociation of $H_2SO_4$ under dry condition and influence of organics/SOAs on DRH of AS need to be discussed.

There are several places in the paper are inconsistent and some explanations are a bit too obscure as detailed in the major comments below. Beyond these, I do not see any major obstacles to publication.

**Major comments:**

1. Line 230

For all the experiments in Figure 2, when all the limonene was consumed, the particle volume concentration seemed kept increasing, *i.e.,* Exp.#7, dark blue markers vertically stacked at 30 ppb limonene. Does that mean that particle volume concentration kept increasing after limonene was completely consumed? This is inconsistent with Figure 1 that when all the limonene was consumed, the particle volume concentration reached a plateau.

2. Line 235 & Line 245

The authors attribute the increased SOA formation to increased particle acidity.

Experiments #1 - #10 were performed at RH below 16 %, where the particles might have less water. I'm not sure particle acidity is increased as it is a reflection of hydrogen activity, where aerosol water is needed.  See comments earlier.

3. Line 250

The authors stated that "increase in particle volume concentration was also observed in the presence of $SO_2$ under humid conditions.

In Figure 1, I cannot tell if the increase is by comparing Seed volume and Final volume between Exp #11-13 and Exp #14-15, please clarify.

4. Line 255

The authors stated that aerosol water decreased particle acidity, resulting in decreased SOA enhancement in humid condition. However, if we compare experiments without $SO_2$, i.e., Expt #1-3,  with Expt #11-13, SOA enhancement also showed a decrease under humid condition when compared with dry condition. Acidity should not be the issue in the experiments without $SO_2$, does that mean it is the humidity that affects the SOA yield instead of the acid catalyzed formation?

5. Line 270

$SO_2$ can dissolve into the aqueous droplets, what is its contribution to the measured SO2 gas consumption?

6. Lin 320 & Figure 4

I understand the authors conducted flow tube experiments in order to collect more products for analysis. But I'm not sure the flow tube experiments are exactly comparable to the chamber ones. The key issue is that OH scavenger is not added in the flow tube experiments. It is more efficient in producing peroxides in OH system than in $O_3$ system due to the RO2 + HO2 reactions. Therefore, the bulk experiments using the LSOA extracts without scavenger will bias the absorption of $SO_2$ high compared with the real chamber SOA material where OH scavenger is present.

7. Line 300

Even $H_2SO_4$ can be formed, I'm not sure in terms of the fate of "Sulfur", which is formed more significantly $H_2SO_4$ and OS? This is related to the mechanism/yield of SOA formation under dry condition. The authors stated aerosol acidity increased SOA yield but what is the acidity of aerosols in dry condition? And the authors did observe quite some OS molecules, will this be the major reason for increased SOA yield? Please clarify.

8. Line 345

As the author said, aqueous reaction of $SO_2$ with $O_3$ in forming sulfate is proved to be quite efficient. What is the reason that $SO_2$ is not reacted efficiently with $O_3$ in these experiments? Any estimates for the rate constant of $SO_2+O_3$ in this experiment?

9. Line 370

What is the resolution of the mass spectrometers? I wonder if it can give four decimals for the detected m/z.

10. Line 420

"first generation of oxidation products from alpha-pinene ozonolysis may be too volatile to condense…"

There are quite some studies (*i.e.*, Ehn et al., Nature, 2014) showing that first generation of oxidation products were supposed to be HOMs, which are of extremely low vapor pressures and can condense easily. Please explain.

11. Line 455

"We present evidence to suggest that $HSO_3^-$ can further react with organic peroxides produced from monoterpene ozonolysis"

I don't think the authors give any explanation before line 455 about reaction between $HSO_3^-$ with organic peroxides. The authors only show the evidence that organic peroxides decreased when bubbling $SO_2$ into the solution. Please explain the reaction mechanism of $HSO_3^-$ with organic peroxides.

**Reference:**

Ehn, M.; Thornton, J. A.; Kleist, E.; Sipila, M.; Junninen, H.; Pullinen, I.; Springer, M.; Rubach, F.; Tillmann, R.; Lee, B.; *et al.* A Large Source of Low-Volatility Secondary Organic Aerosol. *Nature* **2014**, *506*, 476–479.

---

## Author Comment (AC1) · 20 Feb 2018

**Response to Reviewer 1's comments:**

We thank the reviewer for the comments. Below are our responses to the comments and corresponding modifications to the manuscript.

1. Experiments were made in presence of excess ozone to ensure the complete consumption of the terpenes. Have you performed tests experiments where this was not the case? Would expect any difference in the chemical regimes to $SO_2$ oxidation at lower ozone concentration? For instance, due to reduced peroxy-functions productions, concomitant presence of unsaturations, etc:

Yes, we have performed chamber experiments with limited ozone injection, as shown in Fig. R1. Ozone was injected into the chamber that was prefilled with limonene until it reached ~ 50 ppb. Additional ozone was added at 180 min. As can be observed from the figure, $SO_2$ consumption coincides with limonene depletion and particle growth, further confirming the interactions between $SO_2$ and limonene oxidation products. Comparing this result (e.g. the result of the 1st ozone injection) to the experiment conducted under ozone-rich conditions with similar $SO_2$ injection (Exp. #5, Table 1), lower SOA yield (41% vs. 52%) and smaller $SO_2$ consumption (4.0 ppb vs. 5.2 ppb) were observed, consistent with the results from other studies (Chen and Hopke, 2010; Leungsakul et al., 2005; Youssefi and Waring, 2014). This is likely due to the incomplete oxidation of the two double bonds of limonene or high volatility of second-generation reactions under ozone-limited conditions. We have not examined the detailed chemical changes of SOA products between these two scenarios (ozone-rich vs. ozone-limit). However, based on our observations regarding SOA yields and $SO_2$ consumption, as also mentioned by the reviewer, reduced formation of Criegee intermediates and peroxides is expected, which may weaken the effect of $SO_2$ on limonene SOA formation. Unsaturated compounds may also be present when limited ozone was injected (Maksymiuk et al., 2009).

The following content has been added and highlighted in the manuscript (Section 3.1):

"…Tests have also been performed by injecting ozone in two separate batches into the chamber prefilled with limonene, as shown in Fig. S1. Similar to the experiments conducted under ozone-rich conditions (e.g. Fig. 1), synergistic effects have been observed. We therefore infer from the correlation between depletion rate of $SO_2$ and particle formation that similar species or processes are responsible for $SO_2$ reaction and LSOA formation."

[Figure]

**Figure R1** Particle volume concentration, limonene concentration and SO$_2$ concentration as a function of experimental time with stepwise ozone injection. Ozone was injected into the chamber that was prefilled with limonene until it reached around 50 ppb. Additional ozone was added at 180 min.

2. Maybe you could add in the experimental section 2.2, the ozone levels used in the flow tube experiments with, maybe, some indications how this compares to the chamber ones?

Thanks for the comments. The ozone concentration used in the flow tube experiments was around 3 ppm (in the absence of monoterpenes) as mentioned in Section 2.2. The inlet concentration of ozone was 6 times higher than those of monoterpenes. This concentration was chosen to achieve the greatest extent of oxidation in the flow tube. Based on literature rate constants, the lifetimes of α-pinene and limonene were calculated to be around 3 min and 2 min at these ozone concentrations, respectively. The residence time of chemical species in the flow tube reactor was 4 min.

Compared to chamber experiments, the flow residence time in the flow tube reactor is shorter, which may lead to less-oxidized SOA formation in the flow tube. However, this does not affect our conclusions regarding the effects of SO$_2$ on the change of SOA composition. The observations of organosulfate formation and increased SOA oxidation state in the presence of SO$_2$, are still valid despite the difference between the chamber and the flow tube.

The following content has been added into Section 3.4 of the manuscript:

"….It should also be noted that compared to chamber experiments, SOA formed in the flow tube may be less oxidized due to the short residence time. However, this does not affect our conclusions regarding the effects of $SO_2$ on the change of SOA composition. The observations of organosulfate formation and increased SOA oxidation state in the presence of $SO_2$, are still valid despite the differences in residence time between the chamber and the flow tube experiments."

3. You are making clear that limonene is needed to induce a $SO_2$ loss (line 225), but does this fully exclude that loss of $SO_2$ is not firstly physically driven by solubilization into the nascent aerosols? (Adding limonene would also affect such an equilibrium due to the growth rate of the particles).

Dissolution of $SO_2$ into the particles is governed by Henry's law. We calculated the dissolved $SO_2$ in two scenarios (acidic and neutral). pH = 5 is used for the calculation in acidic scenario because the pH for pure ammonium sulfate particle was estimated to be ~5 according to E-AIM Aerosol Thermodynamics Model (Model II, Clegg et al., 1998).

The effective $SO_2$ Henry's law constants $H^*_{S(IV)}$ under these two scenarios were taken to be $1 \times 10^3$ M atm$^{-1}$ (pH = 5) and $2 \times 10^5$ M atm$^{-1}$ (pH = 7), respectively (Seinfeld and Pandis, 2006). Assuming an initial $SO_2$ concentration of 600 ppb (the maximum $SO_2$ concentration used in this study), the concentrations of dissolved sulfur [S(IV)] in the aqueous phase can be calculated as 0.60 mM (pH = 5) and 120 mM (pH = 7). Assuming that the entire particle is aqueous, the aqueous mass concentration would be ~100 μg/m$^3$, and dissolution of $SO_2$ would only result in a decrease of $1.5 \times 10^{-3}$ ppb (pH = 5) and 0.3 ppb (pH = 7) gaseous $SO_2$, which is at least one order of magnitude less than observed. Therefore, we conclude that the loss of $SO_2$ into particle phase solely due to dissolution is negligible.

4. The paragraph starting at line 230 is slightly confusing to me. How do you conclude/affect to the growth effect to sulfuric acid? How would that acid be produced efficiently in your system? OH reaction can be excluded due to the presence of an OH scavenger and ozone reacts quite slowly under acidic conditions. Would all this just be linked to sCI chemistry? Maybe adding a few words of explanation would be helpful.

The formation of sulfuric acid was predicted based on the chemical mass balance. By assuming that all the loss of $SO_2$ leads to the formation of sulfuric acid, we can estimate the upper limit of sulfuric acid that was formed in the condensed phase in our experiments.

Oxidation of $SO_2$ by Criegee intermediates from monoterpene ozonolysis leads to efficient production of sulfuric acid (Sipilä et al., 2014). However, it is also observed in this study that $SO_2$ can be oxidized by organic peroxides in the particle phase, which may also contribute to the formation of sulfuric acid especially under humid conditions. The relative contributions of each pathway under dry and humid conditions are shown in Fig. 8A and 8B.

5. Humidity seems to reduce the SOA enhancement, as measured in the chamber. But what about the products distribution is humidity reducing the amount of organosulfate being produced? Also, as noted, an increased humidity leads to an increased particle phase pH and hence an increased reactivity of ozone toward $SO_2$ leading to sulfate production that may explain partly the observed enhanced $SO_2$ loss under humid conditions. Would that be a sign of a competition between ozone reactivity and peroxy-type chemistry leading to organosulfate production, as this exist for cloud processing of $SO_2$ (which is strongly pH dependent)?

Thanks for the comments. The role of humidity in organosulfate formation can be complicated. On one hand, increased humidity reduces acidity in the particle phase. However, particle acidity was found to promote the formation of organosulfates. Surratt et al. (2008) demonstrated that sulfate formation (including inorganic sulfate and organosulfates) in isoprene and monoterpene oxidation system increased with increasing seed particle acidity. Chan et al. (2011) observed that the abundances of organosulfates from β-caryophyllene photooxidation correlates strongly with aerosol acidity. On the other hand, increased humidity enhances the interactions between $SO_2$ and organic peroxides as shown in this study. Conclusion cannot be drawn without detailed mechanism studies whether organosulfate can be formed through $SO_2$/peroxide reactions. However, study is currently conducted in this group to examine the interactions between organic peroxide and $SO_2$.

With regards to the effect of ozone, we have conducted control experiment by adding $SO_2$, ozone, formic acid and ammonium sulfate into the chamber, as shown in Fig. S4A in the Supporting Information. No significant decrease of $SO_2$ was observed, indicating that dissolved ozone in the aqueous phase is not an important sink of $SO_2$ in this study. This is likely because that ammonium sulfate seed is slightly acidic (pH ~ 5, based on the calculation from E-AIM Aerosol Thermodynamic Model) that is less favorable for $SO_2$ oxidation by ozone in the aqueous phase. In addition, unlike $SO_2$ oxidation in a cloud droplet, the liquid water content in ammonium sulfate particles is limited. Therefore, little $SO_2$ depletion was observed. However, we agree with the reviewer that competition exists between $SO_2$/ozone reaction and $SO_2$/peroxide chemistry under high pH conditions and during cloud processing.

6. Line 76: please add the reference for the statement: "The reaction rate of particle phase $SO_2 + RO_2$" Line 88: add also the reference to: Passananti, M.; Kong, L. D.; Shang, J.; Dupart, Y.; Perrier, S.; Chen, J. M.; Donaldson, D. J.; George, C. Organosulfate Formation through the Heterogeneous Reaction of Sulfur Dioxide with Unsaturated Fatty Acids and Long-Chain Alkenes. Angewandte Chemie-International Edition 2016, 55 (35), 10336-10339.

Thanks for the comments. The references were added into the manuscript.

**Reference:**

Chan, M. N., Surratt, J. D., Chan, A. W. H., Schilling, K., Offenberg, J. H., Lewandowski, M., Edney, E. O., Kleindienst, T. E., Jaoui, M., Edgerton, E. S., Tanner, R. L., Shaw, S. L., Zheng, M., Knipping, E. M. and Seinfeld, J. H.: Influence of aerosol acidity on the chemical composition of secondary organic aerosol from β-caryophyllene, Atmos. Chem. Phys., 11(4), 1735–1751, doi:10.5194/acp-11-1735-2011, 2011.

Chen, X. and Hopke, P. K.: A chamber study of secondary organic aerosol formation by limonene ozonolysis, Indoor Air, 20(4), 320–328, doi:10.1111/j.1600-0668.2010.00656.x, 2010.

Clegg, S. L., Brimblecombe, P. and Wexler, A. S.: Thermodynamic model of the system $H^+-NH_4^+-SO_4^{2-}-NO_3^--H_2O$ at tropospheric temperatures, J. Phys. Chem. A, 102(12), 2137–2154, doi:10.1021/jp973042r, 1998.

Leungsakul, S., Jaoui, M. and Kamens, R. M.: Kinetic mechanism for predicting secondary organic aerosol formation from the reaction of d-limonene with ozone, Environ. Sci. Technol., 39(24), 9583–9594, doi:10.1021/es0492687, 2005.

Maksymiuk, C. S., Gayahtri, C., Gil, R. R. and Donahue, N. M.: Secondary organic aerosol formation from multiphase oxidation of limonene by ozone: mechanistic constraints via two-dimensional heteronuclear NMR spectroscopy, Phys. Chem. Chem. Phys., 11(36), 7810–7818, doi:10.1039/B820005J, 2009.

Seinfeld, J. H. and Pandis, S. N.: Atmospheric chemistry and physics: from air pollution to climate change, 2nd Ed., Wiley: New York., 2006.

Sipilä, M., Jokinen, T., Berndt, T., Richters, S., Makkonen, R., Donahue, N. M., Mauldin III, R. L., Kurtén, T., Paasonen, P., Sarnela, N., Ehn, M., Junninen, H., Rissanen, M. P., Thornton, J., Stratmann, F., Herrmann, H., Worsnop, D. R., Kulmala, M., Kerminen, V.-M. and Petäjä, T.: Reactivity of stabilized Criegee intermediates (sCIs) from isoprene and monoterpene ozonolysis toward $SO_2$ and organic acids, Atmos. Chem. Phys., 14(22), 12143–12153, doi:10.5194/acp-14-12143-2014, 2014.

Surratt, J. D., Gómez-González, Y., Chan, A. W. H., Vermeylen, R., Shahgholi, M., Kleindienst, T. E., Edney, E. O., Offenberg, J. H., Lewandowski, M., Jaoui, M., Maenhaut, W., Claeys, M., Flagan, R. C. and Seinfeld, J. H.: Organosulfate formation in biogenic secondary organic aerosol, J. Phys. Chem. A, 112(36), 8345–8378, doi:10.1021/jp802310p, 2008.

Youssefi, S. and Waring, M. S.: Transient secondary organic aerosol formation from limonene ozonolysis in indoor environments: Impacts of air exchange rates and initial concentration ratios, Environ. Sci. Technol., 48(14), 7899–7908, doi:10.1021/es5009906, 2014.

---

## Author Comment (AC2) · 20 Feb 2018

**Response to Reviewer 2's comments:**

We thank the reviewer for the comments. Below are our responses to the comments and corresponding modifications to the manuscript.

1. OS vs. inorganic sulfate. OS and oligomers analyses are not quantitative, it is not easy to rely on their peak identifications to explain the difference in yields under wet and dry conditions.

We agree with the reviewer that the organosulfate and oligomer analyses based on the ESI mass spectra may not be quantitative. As mentioned in the manuscript, the negative mode in ESI is sensitive only to acidic species. Therefore, the effects of  $SO_2$  on relative signal fractions and oxidation states observed in this work may only be valid for these species. Though we observed increased total signal fraction of organosulfate ions with increased  $SO_2$  concentration, since no authentic standards were available for quantification, no conclusions can be drawn about the difference in organosulfate amounts between the two experiments.

However, our qualitative analyses by linking the ion drift time and the Kendrick mass defect to the molecular formula of each peak in the ESI spectra highlights the formation of organosulfates. And the number of the organosulfates that were identified increased with increasing  $SO_2$  injection.

2. Aerosol acidity has been used heavily to explain the results. Need better, at least, semiquantitative discussions, on aerosol acidity. Dissociation of  $H_2SO_4$  under dry condition and influence of organics/SOAs on DRH of AS need to be discussed.

For aerosol acidity, please refer to the response to Comment 4 and Comment 7.

With regards to the influence of  $\alpha$ -pinene and limonene SOA on the hygroscopicity of ammonium sulfate particles, Takahama et al. (2007) observed that SOA from  $\alpha$ -pinene and limonene ozonolysis has negligible effect on the efflorescence transitions of ammonium sulfate particles. Smith et al. (2011) demonstrated that both the deliquescence and efflorescence of ammonium sulfate were minimally affected by SOA from  $\alpha$ -pinene ozonolysis.

As discussed in Section 2.1 in the manuscript, under dry conditions, ammonium sulfate particles were generated from atomizer, followed by drying with silica gel diffusion dryers. The RH used in the experiments under dry conditions (~10%) were below the efflorescence relative humidity of ammonium sulfate. Therefore, we expect that particles remain in a solid phase with limited water content present. However, under humid conditions (~50%), no diffusion dryer was used before injecting ammonium sulfate particles, suggesting that those particles are in a liquid form with higher liquid water content compared to those seeds under dry conditions.

**3. Line 230**

For all the experiments in Figure 2, when all the limonene was consumed, the particle volume concentration seemed kept increasing, i.e., Exp.#7, dark blue markers vertically stacked at 30 ppb limonene. Does that mean that particle volume concentration kept increasing after limonene was completely consumed? This is inconsistent with Figure 1 that when all the limonene was consumed, the particle volume concentration reached a plateau.

Thanks for the comment. The particle volume concentration continued to increase even when the majority of limonene was consumed. Fig. 1R plots the formation of particles as a function of limonene consumption shown in Fig. 1 (during the first limonene injection). Increased particle formation was observed even when the consumption of limonene was close to completion. This continued increase of particle mass concentration was also observed in studies by Ng et al. (2006) and Zhang et al. (2006) and was attributed to the multi-generation (first- and second-generation) oxidation and significantly slower (ratelimiting) oxidation of the exocyclic double bond in limonene. However, it should be noted that both the first- and the second-generation reactions took place over the course of the experiment. The second-generation reaction of limonene SOA did not solely exist after limonene was completely consumed (i.e. after 30 ppb shown in Fig. 1R). The enhancement of particle volume concentration after the complete consumption of limonene varied slightly from experiment to experiment due to factors such as chamber mixing, and/or particle wall loss correction and/or varied experimental conditions. However, this difference could be minimized by calculating the overall SOA yields. As shown in Table 1, the overall SOA yields increased with increasing SO2 concentrations.

**Figure R1** The relationship between particle volume concentration and limonene consumption shown in Fig. 1 (for the first limonene injection only). Increased particle volume concentration was observed even when limonene was completely consumed.

**4. Line 235 & Line 245**

The authors attribute the increased SOA formation to increased particle acidity. Experiments #1 - #10 were performed at RH below 16 %, where the particles might have less water. I'm not sure particle acidity is increased as it is a reflection of hydrogen activity, where aerosol water is needed. See comments earlier.

Enhanced SOA formation have been observed in the presence of acidic seeds under dry conditions (RH  $\leq 20\%$ ) (Czoschke et al., 2003; Czoschke and Jang, 2006; Jang et al., 2002; Northcross and Jang, 2007). SOA yields from monoterpene ozonolysis were found to increase with increasing particle acidity.

We agree with the reviewer that under dry conditions, the particles might have less water. However, it should be noted that sulfuric acid is highly hydroscopic and it takes up water even at very low RH (Biskos et al., 2009; Seinfeld and Pandis, 2006). Condensation of sulfuric acid increases the liquid water content in the particle phase. This amount of water helps the dissociation of sulfuric acid and increases hydrogen ion activity in the particle phase, which enables acid-catalyzed reactions. Under humid conditions, though increased liquid water content may potentially increase the dissociation of sulfuric acid, it may also dilute the molarity of hydrogen ion in the particle phase. As for the hydrogen ion activity and the pH value under dry conditions, please refer to Comment 7 for detailed discussion.

**5. Line 270**

SO2 can dissolve into the aqueous droplets, what is its contribution to the measured SO2 gas consumption?

Dissolution of SO2 into the particles is governed by Henry's law. We calculated the dissolved SO2 in two scenarios (acidic and neutral). pH = 5 is used for the calculation in acidic scenario because the pH for pure ammonium sulfate particle was estimated to be ~5 according to E-AIM Aerosol Thermodynamics Model (Model II, Clegg et al., 1998).

The effective SO2 Henry's law constants  $H_{s(IV)}^*$  under these two scenarios are taken to be  $1 \times 10^3$  M atm-1 (pH = 5) and  $2 \times 10^5$  M atm-1 (pH = 7), respectively (Seinfeld and Pandis, 2006). Assuming an initial SO2 concentration of 600 ppb (the maximum SO2 concentration used in this study), the concentrations of dissolved sulfur [S(IV)] in the aqueous phase can be calculated as 0.60 mM (pH = 5) and 120 mM (pH = 7). Assuming that the entire particle is aqueous, the aqueous mass concentration would be ~100 µg/m3, and dissolution of SO2 would only result in a decrease of  $1.5 \times 10^{-3}$  ppb (pH = 5) and 0.3 ppb (pH = 7) gaseous SO2, which is at least one order of magnitude less than observed. Therefore, we conclude that the loss of SO2 into particle phase solely due to dissolution is negligible.

6. I understand the authors conducted flow tube experiments in order to collect more products for analysis. But I'm not sure the flow tube experiments are exactly comparable to the chamber ones. The key issue is that OH scavenger is not added in the flow tube experiments. It is more efficient in producing peroxides in OH system than in  $O_3$  system due to the  $RO_2 + HO_2$  reactions. Therefore, the bulk experiments using the LSOA extracts without scavenger will bias the absorption of  $SO_2$  high compared with the real chamber SOA material where OH scavenger is present.

Thanks for the comment. OH scavenger was used in all the experiments, both in the flow tube and the chamber experiments. Therefore, the effect of OH on SOA formation in the flow tube was minimal. We apologize for not mentioning it clearly in the manuscript.

We have added the following content into Section 2.2:

"To collect sufficient SOA mass for offline chemical analysis, SOA was also produced in a quartz flow tube by reacting limonene or  $\alpha$ -pinene with ozone (~3 ppm) in the presence or absence of SO2 under dry (10-13% RH) and humid (55-60% RH) conditions. The flow tube has a diameter of 10.2 cm and length of 120 cm, and the residence time in the flow tube is 4 min. Cyclohexane was used as OH scavenger. Limonene/cyclohexane (1:1500 v/v) and  $\alpha$ -pinene/cyclohexane solution (1:500 v/v) was prefilled in a 1mL syringe (Hamilton) and injected into the flow tube using a syringe pump (Legato 100, KDS)."

**7. Line 300**

Even  $H_2SO_4$  can be formed, I'm not sure in terms of the fate of "Sulfur", which is formed more significantly  $H_2SO_4$  and OS? This is related to the mechanism/yield of SOA formation under dry condition. The authors stated aerosol acidity increased SOA yield but what is the acidity of aerosols in dry condition? And the authors did observe quite some OS molecules, will this be the major reason for increased SOA yield? Please clarify.

Thanks for the comments. In this study, the formation of both sulfuric acid and organosulfates were observed, as displayed in the ESI spectra in Fig. S5 and the peak assignments of sulfur-containing ions in Table S1. For example, m/z 96.96 (HSO4-) and m/z 194.93 (H2SO4 · HSO4-) show the evidence of the formation of sulfuric acid. This observation is consistent with the results from Sipilä et al. (2014) who also observed efficient sulfuric acid formation from monoterpene ozonolysis with SO2.

With regards to the contribution of organosulfates to SOA mass, Iinuma et al. (2007) demonstrated that organosulfates produced from limonene ozonolysis under acidic conditions contribute at least as much as the first- and second-generation oxidation products to SOA mass. However, to what extent organosulfates contribute to the total particle-phase sulfur and SOA mass in this study is unknown without authentic standards. Further study is in progress in this group to quantify organosulfate formation using synthetic standards.

In terms of aerosol acidity under dry conditions in this study, we did a simple estimation taking Exp. #6 as an example (both of  $SO_2$  injection concentration and seed particle loading in Exp. #6 are in the middle range of all the experiments). Assuming all the consumed  $SO_2$  resulting in sulfuric acid formation in the particle phase, the formed sulfuric acid in the particle phase is calculated to be 0.24 µmol/m3. Ammonium sulfate seed concentration in Exp. #6 is 0.88 µmol/m3 with a density of 1.77 g/cm3. Based on the results from E-AIM Aerosol Thermodynamics Model (Model II; Clegg et al., 1998), the pH was then calculated using the following equations without considering the partitioning of NH3 between the gas phase and the aqueous phase (Clegg et al., 1998):

 $pH = -\log \left[M_{H^+} \cdot \gamma_{H^+}\right]$

where  $M_{H^+}$  and  $\gamma_{H^+}$  are the molarity and molarity-based activity coefficient of hydrogen ion in the aqueous phase, respectively.

Without taking the contribution of organic acids into account, the pH value was estimated to be  $\sim$ 1.2 at 10% RH, which is very acidic. And it is noted that if the partitioning of NH3 is considered, the pH value will be even lower (i.e., even higher acidity).

**8. Line 345**

As the author said, aqueous reaction of  $SO_2$  with  $O_3$  in forming sulfate is proved to be quite efficient. What is the reason that  $SO_2$  is not reacted efficiently with  $O_3$  in these experiments? Any estimates for the rate constant of  $SO_2+O_3$  in this experiment?

With regards to the effect of ozone, we have conducted control experiment by adding  $SO_2$ , ozone, formic acid and ammonium sulfate into the chamber, as shown in Fig. S4A in the Supporting Information. No significant decrease of  $SO_2$  was observed, indicating that dissolved ozone in the aqueous phase is not an important sink of  $SO_2$  under the conditions in this study. This is likely because that ammonium sulfate seed is slightly acidic (pH ~ 5) that is less favorable for  $SO_2$  oxidation by ozone in the aqueous phase. According to Seinfeld and Pandis (2006), the rate of  $SO_2$  and ozone reaction in the aqueous phase at pH = 4-5 is at the level of  $10^6 \text{ M}^{-1} \text{ s}^{-1}$ . In addition, unlike  $SO_2$  oxidation in a cloud droplet, the liquid water content in ammonium sulfate particles is limited. Therefore, little  $SO_2$  depletion was observed.

The following content has been modified into Section 3.3.3:

"...suggesting that reactions between SO2 and ozone either in the gas or particle phase have negligible effects on SO2 consumption. This is likely because that ammonium sulfate seed is slightly acidic. The pH value calculated based on the E-AIM Aerosol Thermodynamic Model for ammonium sulfate at 50% RH is ~5 (Clegg et al., 1998; Wexler and Clegg, 2002). Under this pH condition, SO2 oxidation by ozone is slow and less favorable in the aqueous phase. It is noted that the pH value was estimated without considering the partitioning of trace gases (i.e. NH3). Even lower pH (i.e. higher acidity) would be yielded if taking into account this effect. In addition, unlike SO2 oxidation in a cloud droplet, the liquid water content in ammonium sulfate particles is limited. Therefore, little SO2 depletion was observed."

9. Line 370

What is the resolution of the mass spectrometers? I wonder if it can give four decimals for the detected m/z.

The resolution of the time-of-flight mass spectrometer is around 3500–4000 FWHM at m/z 250. When performing mass calibration, the accuracy of each calibrated mass and the overall calibration was constrained within 5 ppm. For example, if an ion of calculated m/z 250 is observed at m/z 250.001, the mass accuracy is 4 ppm. Therefore, the mass detected by our instrument can be accurate to the third decimal place, with the fourth decimal point estimated.

This information of the mass spectral resolution and mass accuracy has been updated in the manuscript in Section 2.3:

"Particle composition was analyzed using electrospray ionization-ion mobility spectrometry-high resolution time-of-flight mass spectrometry (ESI-IMS-ToF, TOFWERK, hereafter referred to as IMS-TOF) with a mass spectral resolution of 3500-4000 FWHM at m/z 250. Mass calibration was performed before each measurement with a mass accuracy within 5 ppm of each calibration chemical. Details of the IMS-TOF technique are described in recent publications by Krechmer et al. (2016) and Zhang et al., (2016)."

**10. Line 420**

"first generation of oxidation products from alpha-pinene ozonolysis may be too volatile to condense..."

There are quite some studies (i.e., Ehn et al., Nature, 2014) showing that first generation of oxidation products were supposed to be HOMs, which are of extremely low vapor pressures and can condense easily. Please explain.

Thanks for the comment.

We have corrected the statements in the manuscript (Section 3.6):

"Limonene has two double bonds. If SO2 prevents oligomerization of the first-generation products, these products can still react further with ozone to add another oxidized functional groups to form condensable products. As a comparison,  $\alpha$ -pinene only have one double bond. The presence of SO2 reduces oligomerization and limits enhancements in SOA yields."

**11. Line 455**

"We present evidence to suggest that HSO3- can further react with organic peroxides produced from monoterpene ozonolysis" I don't think the authors give any explanation before line 455 about reaction between HSO3- with organic peroxides. The authors only show the evidence that organic peroxides decreased when bubbling SO2 into the solution. Please explain the reaction mechanism of HSO3- with organic peroxides.

Thanks for the comment. It is well known that dissolved  $SO_2$  can be oxidized by peroxides (e.g.,  $H_2O_2$ , methylhydroxyperoxide and peroxyacetic acid) to form sulfate (Lind et al., 1987), as also shown in the following equations:

$$\begin{split} \mathrm{HSO}_{3^{-}} &+ \mathrm{H}_{2}\mathrm{O}_{2} &+ \mathrm{H}^{+} \longrightarrow \mathrm{SO}_{4}^{2^{-}} &+ 2\mathrm{H}^{+} &+ \mathrm{H}_{2}\mathrm{O} \\ \mathrm{HSO}_{3^{-}} &+ \mathrm{CH}_{3}\mathrm{OOH} &+ \mathrm{H}^{+} \longrightarrow \mathrm{SO}_{4}^{2^{-}} &+ 2\mathrm{H}^{+} &+ \mathrm{CH}_{3}\mathrm{OH} \\ \mathrm{HSO}_{3^{-}} &+ \mathrm{CH}_{3}\mathrm{C}(\mathrm{O})\mathrm{OOH} &+ \mathrm{H}^{+} \longrightarrow \mathrm{SO}_{4}^{2^{-}} &+ 2\mathrm{H}^{+} &+ \mathrm{CH}_{3}\mathrm{COOH} \end{split}$$

The bubbling experiments conducted in this study have shown significant depletion of the total peroxide content when bubbling  $SO_2$  into either limonene SOA extract or pure organic peroxide, suggesting that peroxides in SOA are reactive to dissolved  $SO_2$ . This is consistent with our experimental observations that greater  $SO_2$  depletion was observed under humid conditions where more dissolved  $SO_2$  was present. Further investigation is undertaken in this group to elucidate the mechanisms and kinetics of  $SO_2$ /peroxide reaction.

**Reference**

Biskos, G., Buseck, P. R. and Martin, S. T.: Hygroscopic growth of nucleation-mode acidic sulfate particles, J. Aerosol Sci., 40(4), 338–347, doi:10.1016/j.jaerosci.2008.12.003, 2009.

Clegg, S. L., Brimblecombe, P. and Wexler, A. S.: Thermodynamic model of the system  $H^+-NH_4^+-SO_4^{2-}-NO_3^--H_2O$  at tropospheric temperatures, J. Phys. Chem. A, 102(12), 2137–2154, doi:10.1021/jp973042r, 1998.

Czoschke, N. M. and Jang, M.: Acidity effects on the formation of  $\alpha$ -pinene ozone SOA in the presence of inorganic seed, Atmos. Environ., 40(23), 4370–4380, doi:10.1016/j.atmosenv.2006.03.030, 2006.

Czoschke, N. M., Jang, M. and Kamens, R. M.: Effect of acidic seed on biogenic secondary organic aerosol growth, Atmos. Environ., 37(30), 4287–4299, doi:10.1016/S1352-2310(03)00511-9, 2003.

Iinuma, Y., Müller, C., Böge, O., Gnauk, T. and Herrmann, H.: The formation of organic sulfate esters in the limonene ozonolysis secondary organic aerosol (SOA) under acidic conditions, Atmos. Environ., 41(27), 5571–5583, doi:10.1016/j.atmosenv.2007.03.007,

2007.

Jang, M., Czoschke, N. M., Lee, S. and Kamens, R. M.: Heterogeneous atmospheric aerosol production by acid-catalyzed particle-phase reactions, Science, 298(5594), 814–817, doi:10.1126/science.1075798, 2002.

Lind, J. A., Lazrus, A. L. and Kok, G. L.: Aqueous phase oxidation of sulfur(IV) by hydrogen peroxide, methylhydroperoxide, and peroxyacetic Acid, J. Geophys. Res., 92(D4), 4171–4177, doi:10.1029/JD092iD04p04171, 1987.

Ng, N. L., Kroll, J. H., Keywood, M. D., Bahreini, R., Varutbangkul, V., Flagan, R. C., Seinfeld, J. H., Lee, A. and Goldstein, A. H.: Contribution of first- versus second-generation products to secondary organic aerosols formed in the oxidation of biogenic hydrocarbons, Environ. Sci. Technol., 40(7), 2283–2297, doi:10.1021/es052269u, 2006.

Northcross, A. L. and Jang, M.: Heterogeneous SOA yield from ozonolysis of monoterpenes in the presence of inorganic acid, Atmos. Environ., 41(7), 1483–1493, doi:10.1016/j.atmosenv.2006.10.009, 2007.

Seinfeld, J. H. and Pandis, S. N.: Atmospheric chemistry and physics: from air pollution to climate change, 2nd Ed., Wiley: New York., 2006.

Sipilä, M., Jokinen, T., Berndt, T., Richters, S., Makkonen, R., Donahue, N. M., Mauldin III, R. L., Kurtén, T., Paasonen, P., Sarnela, N., Ehn, M., Junninen, H., Rissanen, M. P., Thornton, J., Stratmann, F., Herrmann, H., Worsnop, D. R., Kulmala, M., Kerminen, V.-M. and Petäjä, T.: Reactivity of stabilized Criegee intermediates (sCIs) from isoprene and monoterpene ozonolysis toward SO2 and organic acids, Atmos. Chem. Phys., 14(22), 12143–12153, doi:10.5194/acp-14-12143-2014, 2014.

Smith, M. L., Kuwata, M. and Martin, S. T.: Secondary organic material produced by the dark ozonolysis of  $\alpha$ -pinene minimally affects the deliquescence and efflorescence of ammonium sulfate, Aerosol Sci. Technol., 45(2), 244–261, doi:10.1080/02786826.2010.532178, 2011.

Takahama, S., Pathak, R. K. and Pandis, S. N.: Efflorescence transitions of ammonium sulfate particles coated with secondary organic aerosol, Environ. Sci. Technol., 41(7), 2289–2295, doi:10.1021/es0619915, 2007.

Zhang, J., Huff Hartz, K. E., Pandis, S. N. and Donahue, N. M.: Secondary organic aerosol formation from limonene ozonolysis: Homogeneous and heterogeneous influences as a function of NOx, J. Phys. Chem. A, 110(38), 11053–11063, doi:10.1021/jp062836f, 2006.